# A Causal Analysis of Harm

**Sander Beckers**
Cluster of Excellence in Machine Learning
University of Tübingen and
Munich Center for Mathematical Philosophy, LMU
srekcebrednas@gmail.com

**Hana Chockler**
causaLens and
Department of Informatics
King's College London
hana.chockler@kcl.ac.uk

**Joseph Y. Halpern**
Computer Science Department
Cornell University
halpern@cs.cornell.edu

## Abstract

As autonomous systems rapidly become ubiquitous, there is a growing need for a legal and regulatory framework that addresses when and how such a system harms someone. There have been several attempts within the philosophy literature to define harm, but none of them has proven capable of dealing with the many examples that have been presented, leading some to suggest that the notion of harm should be abandoned and "replaced by more well-behaved notions". As harm is generally something that is caused, most of these definitions have involved causality at some level. Yet surprisingly, none of them makes use of causal models and the definitions of actual causality that they can express. In this paper we formally define a qualitative notion of harm that uses causal models and is based on a well-known definition of actual causality [13]. The key features of our definition are that it is based on *contrastive* causation and uses a default utility to which the utility of actual outcomes is compared. We show that our definition is able to handle the examples from the literature, and illustrate its importance for reasoning about situations involving autonomous systems.

## 1 Introduction

The notion that one should not cause harm is a central tenet in many religions; it is enshrined in the medical profession's Hippocratic Oath, which states explicitly "I will do no harm or injustice to [my patients]" [20] it is also a critical element in the law. Not surprisingly, there have been many attempts in the philosophy literature to define harm. Motivated by the observation that we speak of "causing harm", most of these have involved causality at some level. All these attempts have encountered difficulties. Indeed, Bradley [4] says:

*Unfortunately, when we look at attempts to explain the nature of harm, we find a mess. The most widely discussed account, the comparative account, faces counterexamples that seem fatal. But no alternative account has gained any currency. My diagnosis is that the notion of harm is a Frankensteinian jumble . . . It should be replaced by other more well-behaved notions.*

The situation has not improved much since Bradley's paper (see, e.g., recent accounts like [5, 8]). Yet the legal and regulatory aspects of harm are becoming particularly important now, as autonomous systems become increasingly more prevalent. In fact, the new proposal for Europe's AI act [7] contains over 25 references to "harm" or "harmful", saying such things as ". . . it is appropriate to classify [AI systems] as high-risk if, in the light of their intended purpose, they pose a high risk of

36th Conference on Neural Information Processing Systems (NeurIPS 2022).

harm to the health and safety or the fundamental rights of persons ..." [7, Proposal preamble, clause (32)]. Moreover, the European Commission recognized that if harm is to play such a crucial role, it must be defined carefully, saying "Stakeholders also highlighted that ... it is important to define ... 'harm' [7, Part 2, Section 3.1]. Legislative bodies in the UK are also discussing the question of harm and who caused harm in the case of accidents involving autonomous vehicles. The Law Commission of England and Wales and the Scottish Law Commission are recommending that drivers of self-driving cars should not be legally responsible for crashes; rather, the onus should lie with the manufacturer [6]. In particular, if there is harm then this is caused by the manufacturers. The manufacturers translate this recommendation to a standard according to which the driver does not even have to pay attention while at the wheel. If a complex situation arises on the road requiring the driver's attention, the car will notify the driver, giving them 10 seconds to take control. If the driver does not react in time, the car will flash emergency lights, slow down, and eventually stop [18]. Consider the following example (to which we return later).

**Example 1 (Autonomous Car)**  An autonomous car detects an unexpected stationary car in front of it on a highway. It could alert the driver Bob, who would then have to react within 10 seconds. However, 10 seconds is too long: the car will crash into the stationary car within 8 seconds. The autonomous car's algorithm directs it to crash into the safety fence on the side of the highway, injuring Bob. Bob claims that he was harmed by the car. Moreover, he also claims that, if alerted, he would have been able to find a better solution that would not have resulted in his being injured (e.g., swerving into the incoming traffic then back to his own lane after passing the stationary car). We assume that if the autonomous car had done nothing and collided with the stationary car, both drivers would have been injured much more severely.                                                                           □

While the causal model depicting this story is fairly straightforward, the decision on whether harm was caused to Bob, and if yes, who or what caused the harm, is far less clear. Indeed, the philosophy literature seems to suggest that trying to determine this systematically is a lost cause. But as this example illustrates, the stakes of having a well-defined notion of harm have become much higher with the advent of automated decision-making. In contrast to human agents, such systems do not have an informal understanding of harm that informs their actions; so we need a formal definition. Situations like that described in Example 1 are bound to arise frequently in the interaction of autonomous systems with human users, in a variety of domains. We briefly outline two of those.

Imagine a UAV used by the military has to decide whether or not it should bomb a suspected enemy encampment. The problem is that the target is not clearly identified, because there are two camps close to each other: one consisting of civilian refugees, another consisting of a rebel group that is about to launch a deadly attack on the refugee camp, killing all of its inhabitants. The UAV's decision is based only on the expected utility of the refugees, and therefore it bombs the camp. Tragically, as it turns out, the camp was that of the refugees. Here we have the intuition that the UAV harmed these refugees, despite the fact that both actions would have led to all the refugees being killed. Examples in which one event (the bombing) preempts another event (the attack) from causing an outcome are known as *Late Preemption* examples in the causality literature; we discuss them later in the paper.

In the healthcare domain, autonomous systems are used for, among other things, classifying MRI brain images suspected of containing a tumor. If an image is classified as having a tumor, the system decides whether to recommend a surgery. While the overall accuracy of the system is superior to that of humans, in some instances the system overlooks an operable tumor. Imagine a patient who has such a tumor and dies from brain cancer as the result of not undergoing surgery, leading to a dispute between the patient's family and the hospital regarding whether the patient was harmed. Even if both parties agree that the patient would probably have been alive if the diagnosis had been performed by a human, the hospital might claim that using the system is the optimal policy, and therefore one should compare the actual outcome only to those that could have occurred under the policy.

Fortunately, the formal tools at our disposal to develop a formal notion of harm have also improved over the past few years; we take full advantage of these developments in this paper. Concretely, we provide a formal definition of harm that we believe deals with all the concerns that have been raised, seems to match our intuitions well, and connects closely to work on decision theory and utility. Here we briefly give a high-level overview of the key features of our approach and how they deal with the problems raised in the earlier papers.

There is one set of problems that arise from using counterfactuals that also arise with causality, and can be dealt with using the by-now standard approaches in defining causality. For example, Carlson,

Johansson, and Risberg [5] raise a number of problems with defining harm causally that are solved by simply applying the definition of actual causality given by Halpern [12, 13]. The issue of whether failing to take an action can be viewed as causing harm (e.g., can failing to water a neighbor's plants after promising to do so be viewed as causing harm) can also be dealt with by using the standard definition of causality (which allows lack of an action to be a cause).

We remark that Richens, Beard, and Thompson [22] (RBT from now on) also recently observed that using causality appropriately could deal with some of the problems raised in the harm literature.[1] They also give a formal definition of harm that uses causal models, but it does not make use of a sophisticated definition of actual causality such as the one given by Halpern [12, 13]. (See Section 5 for a comparison of our approach to theirs and more discussion of this issue.) RBT focus on the more quantitative, probabilistic aspects of harm. We also believe that a quantitative account is extremely important; we offer such an account in [2]. Conceptually though, the qualitative notion comes first: only after establishing whether or not there was harm does it make sense to ask how much harm occurred. Indeed, our quantitative account generalizes the qualitative account we develop here in several ways (see Section 6).

In any case, just applying the definition of causality does not deal with all problems. The other key step that we take is to assume that there exists a *default* utility. Roughly speaking, we define an event to cause harm whenever it causes the utility of the outcome to be lower than the default utility. The default may be context-dependent, and there may be disagreement about what the default should be. We view that as a feature of our definition. For example, we can capture the fact that people disagree about whether a doctor euthanizing a patient in great pain causes harm by taking it to be a disagreement about what the appropriate default should be. Likewise, the dispute between the family and the hospital described above can be modeled as a disagreement about the right default. Moreover, by explicitly bringing utility into the picture, we can connect issues that that have been discussed at length regarding utility (e.g., what the appropriate discount factor to apply to the utility of future generations is) to issues of harm.

## 2   Causal Models and Actual Causality

We start with a review of causal models [15], since they play a critical role in our definition of harm. The material in this section is largely taken from [13]. We assume that the world is described in terms of variables and their values. Some variables may have a causal influence on others. This influence is modeled by a set of *structural equations*. It is conceptually useful to split the variables into two sets: the *exogenous* variables, whose values are determined by factors outside the model, and the *endogenous* variables, whose values are ultimately determined by the exogenous variables. The structural equations describe how these values are determined.

Formally, a *causal model* $M$ is a pair $(\mathcal{S}, \mathcal{F})$, where $\mathcal{S}$ is a *signature*, which explicitly lists the endogenous and exogenous variables and characterizes their possible values, and $\mathcal{F}$ defines a set of *(modifiable) structural equations*, relating the values of the variables. A signature $\mathcal{S}$ is a tuple $(\mathcal{U}, \mathcal{V}, \mathcal{R})$, where $\mathcal{U}$ is a set of exogenous variables, $\mathcal{V}$ is a set of endogenous variables, and $\mathcal{R}$ associates with every variable $Y \in \mathcal{U} \cup \mathcal{V}$ a nonempty set $\mathcal{R}(Y)$ of possible values for $Y$ (i.e., the set of values over which $Y$ *ranges*). For simplicity, we assume here that $\mathcal{V}$ is finite, as is $\mathcal{R}(Y)$ for every endogenous variable $Y \in \mathcal{V}$. $\mathcal{F}$ associates with each endogenous variable $X \in \mathcal{V}$ a function denoted $F_X$ (i.e., $F_X = \mathcal{F}(X)$) such that $F_X : (\times_{U \in \mathcal{U}} \mathcal{R}(U)) \times (\times_{Y \in \mathcal{V} - \{X\}} \mathcal{R}(Y)) \to \mathcal{R}(X)$. This mathematical notation just makes precise the fact that $F_X$ determines the value of $X$, given the values of all the other variables in $\mathcal{U} \cup \mathcal{V}$. The structural equations define what happens in the presence of external interventions. Setting the value of some set $\vec{X}$ of variables to $\vec{x}$ in a causal model $M = (\mathcal{S}, \mathcal{F})$ results in a new causal model, denoted $M_{\vec{X} \leftarrow \vec{x}}$, which is identical to $M$, except that the equations for $\vec{X}$ in $\mathcal{F}$ are replaced by $\vec{X} = \vec{x}$.

Note that the causal models we consider here are deterministic. In general, one can also consider *probabilistic causal models*. A probabilistic causal model is a tuple $M = (\mathcal{S}, \mathcal{F}, \Pr)$, where $(\mathcal{S}, \mathcal{F})$ is a causal model, and $\Pr$ is a probability on contexts. Deterministic models suffice for offering a qualitative notion of harm, but we use probabilistic causal models for our quantitative generalization [2].

---

[1] Indeed, a talk by Jonathan Richens that discussed these issues (which are also discussed in [22]) was attended by one of the authors of this paper, and it motivated us to look carefully at harm.

The dependencies between variables in a causal model $M = ((\mathcal{U}, \mathcal{V}, \mathcal{R}), \mathcal{F})$ can be described using a *causal network* (or *causal graph*), whose nodes are labeled by the endogenous and exogenous variables in $M$, with one node for each variable in $\mathcal{U} \cup \mathcal{V}$. The roots of the graph are (labeled by) the exogenous variables. There is a directed edge from variable $X$ to $Y$ if $Y$ *depends on* $X$; this is the case if there is some setting of all the variables in $\mathcal{U} \cup \mathcal{V}$ other than $X$ and $Y$ such that varying the value of $X$ in that setting results in a variation in the value of $Y$; that is, there is a setting $\vec{z}$ of the variables other than $X$ and $Y$ and values $x$ and $x'$ of $X$ such that $F_Y(x, \vec{z}) \neq F_Y(x', \vec{z})$. A causal model $M$ is *recursive* (or *acyclic*) if its causal graph is acyclic. It should be clear that if $M$ is an acyclic causal model, then given a *context*, that is, a setting $\vec{u}$ for the exogenous variables in $\mathcal{U}$, the values of all the other variables are determined (i.e., there is a unique solution to all the equations). We can determine these values by starting at the top of the graph and working our way down. In this paper, following the literature, we restrict to recursive models.

We call a pair $(M, \vec{u})$ consisting of a causal model $M$ and a context $\vec{u}$ a *(causal) setting*. A causal formula $\psi$ is true or false in a setting. We write $(M, \vec{u}) \models \psi$ if the causal formula $\psi$ is true in the setting $(M, \vec{u})$. The $\models$ relation is defined inductively. $(M, \vec{u}) \models X = x$ if the variable $X$ has value $x$ in the unique (since we are dealing with acyclic models) solution to the equations in $M$ in context $\vec{u}$ (that is, the unique vector of values for the exogenous variables that simultaneously satisfies all equations in $M$ with the variables in $\mathcal{U}$ set to $\vec{u}$). Finally, $(M, \vec{u}) \models [\vec{Y} \leftarrow \vec{y}]\varphi$ if $(M_{\vec{Y} \leftarrow \vec{y}}, \vec{u}) \models \varphi$.

A standard use of causal models is to define *actual causation*: that is, what it means for some particular event that occurred to cause another particular event. There have been a number of definitions of actual causation given for acyclic models (e.g., [1, 10, 11, 15, 13, 16, 17, 26, 27]). Although most of what we say in the remainder of the paper applies without change to other definitions of actual causality in causal models, for definiteness, we focus here on what has been called the *modified* Halpern-Pearl definition [12, 13], which we briefly review. (See [13] for more intuition and motivation.)

The events that can be causes are arbitrary conjunctions of primitive events (formulas of the form $X = x$); the events that can be caused are arbitrary Boolean combinations of primitive events. To relate the definition of causality to the (contrastive) definition of harm, we give a contrastive variant of the definition of actual causality; rather than defining what it means for $\vec{X} = \vec{x}$ to be an (actual) cause of $\phi$, we define what it means for $\vec{X} = \vec{x}$ *rather than* $\vec{X} = \vec{x}'$ to be a cause of $\phi$ rather than $\phi'$.

**Definition 1** $\vec{X} = \vec{x}$ *rather than* $\vec{X} = \vec{x}'$ *is an* actual cause *of* $\phi$ *rather than* $\phi'$ *in* $(M, \vec{u})$ *if the following three conditions hold:*

AC1. $(M, \vec{u}) \models (\vec{X} = \vec{x}) \wedge \phi$.

AC2. *There is a set* $\vec{W}$ *of variables in* $\mathcal{V}$ *and a setting* $\vec{w}$ *of the variables in* $\vec{W}$ *such that* $(M, \vec{u}) \models \vec{W} = \vec{w}$ *and* $(M, \vec{u}) \models [\vec{X} \leftarrow \vec{x}', \vec{W} \leftarrow \vec{w}]\phi'$, *where* $\phi' \Rightarrow \neg\phi$ *is valid.*

AC3. $\vec{X}$ *is minimal; there is no strict subset* $\vec{X}''$ *of* $\vec{X}$ *such that* $\vec{X}'' = \vec{x}''$ *can replace* $\vec{X} = \vec{x}'$, *where* $\vec{x}''$ *is the restriction of* $\vec{x}$ *to the variables in* $\vec{X}''$.

AC1 just says that $\vec{X} = \vec{x}$ cannot be considered a cause of $\phi$ unless both $\vec{X} = \vec{x}$ and $\phi$ actually happen. AC3 is a minimality condition, which says that a cause has no irrelevant conjuncts. AC2 captures the standard but-for condition ($\vec{X} = \vec{x}$ rather than $\vec{X} = \vec{x}'$ is a cause of $\phi$ if, had $\vec{X}$ beem $\vec{x}'$ rather than $\vec{x}$, $\phi$ would not have happened) but allows us to apply it while keeping fixed some variables to the value that they had in the actual setting $(M, \vec{u})$. If $\vec{X} = \vec{x}$ is an actual cause of $\phi$ and there are two or more conjuncts in $\vec{X} = \vec{x}$, one of which is $X = x$, then $X = x$ is *part of a cause* of $\phi$. In the special case that $\vec{W} = \emptyset$, we get the standard but-for definition of causality: if $\vec{X} = \vec{x}$ had not occurred (because $\vec{X}$ was $\vec{x}'$ instead) $\phi$ would not have occurred (because it would have been $\phi'$).

The reader can easily verify that $\vec{X} = \vec{x}$ is an actual cause of $\phi$ according to the standard non-contrastive definition [13] iff there exist $\vec{x}'$ and $\phi'$ such that $\vec{X} = \vec{x}$ rather than $\vec{X} = \vec{x}'$ is an actual cause of $\phi$ rather than $\phi'$ according to our contrastive definition.

## 3 Defining Harm

Many definitions of harm have been considered in the literature. The ones most relevant to us are those involving causality and counterfactuals, which have been split into two groups, called the *causal*

*account of harm* and the *counterfactual comparative account account of harm*. Carlson et al. [5] discuss many variants of the causal account; they all have the following structure:

*An event $e$ harms an agent* **ag** *if and only if there is a state of affairs $s$ such that (i) $e$ causes $s$ to obtain, and (ii) $s$ is a harm for* **ag**.

The definitions differ in how they interpret the second clause. We note that although these definitions use the word "cause", it is never defined formally. "Harm" is also not always defined, although in some cases the second clause is replaced by phrases that are intended to be easier to interpret. For example, what Suits [24] calls the *causal-intrinsic badness account* takes $s$ to be a harm for **ag** if $s$ is "intrinsically bad" for **ag**.

The causal-counterfactual account (see, e.g., [9, 19, 25, 9]) has the same structure; the first clause is the same, but now the second clause is replaced by a phrase involving counterfactuals. In its simplest version, this can be formulated as follows: $s$ is a harm for **ag** if and only if **ag** would have been better off had $s$ not obtained.

Even closer to our account is what has been called the *contrastive causal-counterfactual account*. For example, Bontly [3] proposed the following:

*An event $e$ harms a person* **ag** *if and only if there is a state of affairs $s$ and a contrast state of affairs $s'$ such that (i) $e$ rather than a contrast event $e'$ causes $s$ rather than $s'$ to obtain, and (ii)* **ag** *is worse off in $s$ than in $s'$.*

Our formal definition of harm is quite close to Bontly's. We replace "state of affairs" by "outcomes", and associate with each outcome a utility. This is essentially the standard model in decision theory, where actions map states to outcomes, which have associated utilities. Besides allowing us to connect our view to the standard decision-theoretic view (see, e.g., [21, 23]), this choice means that we can benefit from all the work done on utility by decision theorists.

To define harm formally in our framework, we need to both extend and specialize causal models: We specialize causal models by assuming that they include a special endogenous variable $O$ for *outcome*. The various values of the outcome value will be assigned a utility. We often think of an action as affecting many variables, whose values together constitute the outcome. The decision to "package up" all these variables into a single variable $O$ here is deliberate; we do not want to consider the causal impact of some variables that make up the outcome on other variables that make up the outcome (and so do not want to allow interventions on individual variables that make up an outcome; we allow only interventions on complete outcomes). On the other hand, we extend causal models by assigning a utility value to outcomes (i.e., on values of the outcome variable), and by having a default utility.

We thus take a *causal utility model* to be one of the form $M = ((\mathcal{U}, \mathcal{V}, \mathcal{R}), \mathcal{F}, \mathbf{u}, d)$, where $(\mathcal{U}, \mathcal{V}, \mathcal{R}), \mathcal{F}$ is a causal model one of whose endogenous variables is $O$, $\mathbf{u} : \mathcal{R}(O) \rightarrow [0,1]$ is a utility function on outcomes (for simplicity, we assume that utilities are normalized so that the best utility is 1 and the worst utility is 0), and $d \in [0,1]$ is a default utility.[2] As before, we call a pair $(M, \vec{u})$, where now $M$ is a causal utility model and $\vec{u} \in \mathcal{R}(\mathcal{U})$, a setting.

Just like causality, we define harm relative to a setting. Whether or not an event $\vec{X} = \vec{x}$ harms an agent in a given setting will depend very much on the choice of utility function and default value. Thus, to justify a particular ascription of harm, we will have to justify both these choices. In the examples we consider, we typically view the utility function to be **ag**'s utility function, but we are not committed to this choice (e.g., when deciding whether harm is caused by a parent not giving a child ice cream, we may use the parent's definition of utility, rather than the child's one). The choice of a default value is more complicated, and will be discussed when we get to examples; for the definition itself, we assume that we are just given the model, including utility function and default value.

The second clause of our definition is a formalization of Bontly's definition, using the definition of causality given in Section 2, where the events for us, as in standard causal models, have the form $\vec{X} = \vec{x}$ and the alternative events have the form $\vec{X} = \vec{x}'$, and they cause outcomes $O = o$ and $O = o'$, respectively. Unlike Bontly's definition (and others), not only do we require that **ag** is worse off in outcome $o$ (the analogue of state of affairs $s$) than in outcome $o'$ (where "worse off" is formalized by

---

[2]As we said in the introduction, in general, we think of the default utility as being context-dependent, so we really want a function from contexts to default utilities. However, in all the examples we consider in this paper, a single default utility suffices, so for ease of exposition, we make this simplification here.

taking the utility to be lower), we also require the utility of $o$ to be lower than the *default utility*. There is also an issue as to whether we consider there to be harm if $\vec{X} = \vec{x}'$ results in a worse outcome than $o$. Since intuitions may differ here, we formalize this requirement in a third clause, H3, and use it to distinguish between harm and *strict* harm. We will see the effects of our modifications to Bontly's definition when we consider examples in Section 4.

**Definition 2** $\vec{X} = \vec{x}$ harms **ag** in $(M, \vec{u})$, where $M = ((\mathcal{U}, \mathcal{V}, \mathcal{R}), \mathcal{F}, \mathbf{u}, d)$, if there exist $o \in \mathcal{R}(O)$ and $\vec{x}' \in \mathcal{R}(\vec{X})$ such that

H1. $\mathbf{u}(O = o) < d$; and

H2. there exists $o' \in \mathcal{R}(O)$ such that $\vec{X} = \vec{x}$ rather than $\vec{X} = \vec{x}'$ causes $O = o$ rather than $O = o'$ and $\mathbf{u}(O = o) < \mathbf{u}(O = o')$.

$\vec{X} = \vec{x}$ strictly harms **ag** in $(M, \vec{u})$ if, in addition,

H3. $\mathbf{u}(O = o) \leq \mathbf{u}(O = o'')$ for the unique $o'' \in \mathcal{R}(O)$ such that $(M, \vec{u}) \models [\vec{X} \leftarrow \vec{x}'](O = o'')$.

In the special case where Definition 2 is satisfied for some value $o'$ appearing in H2 such that $\mathbf{u}(O = o) < d \leq \mathbf{u}(O = o')$, we say that $\vec{X} = \vec{x}$ causes **ag**'s *utility to be lower than the default*.

It is important to point out that it is quite rare for harm and strict harm to come apart. For one, it requires causation and but-for causation to come apart (otherwise $o' = o''$). In addition, it requires $O$ to have at least three values (otherwise again $o' = o''$). Lastly, even if these two conditions are met, we also need that $\mathbf{u}(O = o'') < \mathbf{u}(O = o) < \mathbf{u}(O = o')$. We discuss one example in the supplementary material (see Section **??**) in which these conditions are met.

As with most concepts in actual causality, deciding whether harm occurred is intractable. Indeed, it is easy to see that it is at least as hard as causality, which is DP-complete [12]. However, this is unlikely to be a problem in practice, since we expect that the causal models that arise when we want to deal with harm will have few variables, which take on few possible values (or will involve many individuals that can all be described with by a small causal model), so we can decide harm by simply checking all possibilities.

It is useful to compare our definition with the counterfactual comparative account of harm. Here it is, translated into our notation:

**Definition 3** $\vec{X} = \vec{x}$ counterfactually harms **ag** in $(M, \vec{u})$, where $M = ((\mathcal{U}, \mathcal{V}, \mathcal{R}), \mathcal{F}, \mathbf{u}, d)$ if there exist $o, o' \in \mathcal{R}(O)$ and $\vec{x}' \in \mathcal{R}(\vec{X})$ such that

C1. $(M, \vec{u}) \models \vec{X} = \vec{x} \land O = o$;

C2. $(M, \vec{u}) \models [\vec{X} \leftarrow \vec{x}'](O = o')$;

C3. $\mathbf{u}(O = o) < \mathbf{u}(O = o')$.

That is, $\vec{X} = \vec{x}$ counterfactually harms **ag** if, for some $x'$ and $o'$, $\vec{X} = \vec{x}$ is what actually happens (C1), $O = o'$ would have happened had $\vec{X}$ been set to $\vec{x}'$ (C2), and **ag** gets higher utility from $o'$ than from $o$ (C3). C1 and C2 together are equivalent to AC1 and AC2 in the special case that $\vec{W} = \emptyset$. That is, C1 and C2 essentially amount to but-for causality. C3 differs from our conditions by not taking into account the default value.

Note that Definition 3 has no analogue of AC3, but all the examples focus on cases where $\vec{X}$ is actually a singleton, so AC3 is trivially satisfied. The key point from our perspective is that the counterfactual comparative account considers only but-for causality, and does not consider a default value. The examples in the next section show how critical these distinctions are.

As mentioned earlier, RBT recently developed a formal account of harm using causal models. While their account is probabilistic and quantitative, we can consider the special case where everything is deterministic and qualitative. When we do this, their account reduces to a strengthening of Definition 3 that brings it somewhat closer to our account: they also suggest using defaults, but have default actions rather than default utilities. In their version of Definition 3, $\vec{X}$ is taken to be the variable representing the action(s) performed and $x'$ is the default action. In order to deal with the limitations of but-for causality, RBT offer a more general account (see their Appendix A) that uses path-specific causality, instead of actual causation. This makes their account different from ours in some significant respects; see Section 5.

# 4 Examples

We now analyse several examples to illustrate how our definition handles the most prominent issues that have been raised in the literature on harm. Bradley [4, p. 398] identifies two such issues that strike him "as very serious", namely the problem of preemption, and the problem of distinguishing harm from merely failing to benefit. These problems therefore serve as a good starting point.

## 4.1 Preemption

To anyone familiar with the literature on actual causation what follows will not come as a surprise. Lewis used examples of preemption to argue that there can be causation without counterfactual dependence (i.e., we need to go beyond but-for causality); this conclusion is now universally accepted. Essentially the same examples show up in the literature on harm: cases of preemption show that an event can cause harm even though the agent's well-being does not counterfactually depend on it. Thus, the counterfactual comparative account of harm fails for the same reason it failed for causality. The good news is that the formal definition of causation (by design) handles problems like preemption well; moreover, the solution carries over directly to our definition of harm. The following vignette is due to Bradley [4], but issues of preemption show up in many papers on causality [1, 11, 15, 13, 17, 26]; all can be dealt with essentially the same way.

**Example 2 (Late Preemption)** Suppose Batman drops dead of a heart attack. A millisecond after his death, his body is hit by a flaming cannonball. The cannonball would have killed Batman if he had still been alive. So the counterfactual account entails that the heart attack was not harmful to Batman. It didn't make things go worse for him. But intuitively, the heart attack was harmful. The fact that he would have been harmed by the flaming cannonball anyway does not seem relevant to whether the heart attack was actually harmful.

In terms of the formal definition, we take $H$ to represent whether Batman has a heart attack ($H = 0$ if he doesn't; $H = 1$ if he does), $C$ to represent if Batman is hit by a cannonball, and $D$ to represent whether Batman dies. Let $\vec{u}$ be the context where $H = 1$. Even without describing the equations, according to the story, $(M, \vec{u}) \models H = 1 \wedge D = 1 \wedge [H = 0](D = 1)$: Batman has a heart attack and he dies, but he would have died even if he did not have a heart attack (since he would have been hit by the cannon ball). Thus, C3 does not hold, since $o = o'$; the outcome is the same whether or not Batman has a heart attack.

The standard causal account handles this problem by introducing two new variables: $K$, for "Batman is killed by the cannonball", and $S$, for "Batman died of a heart attack", to take into account the temporal asymmetry between death due to a heart attack and death due to a cannonball. (We could also deal with this asymmetry by having "time-stamped" variables that talk about when Batman is alive. For more details on incorporating temporal information by using time-stamped variables, see [13].) The causal model has the following equations: $D = S \vee K$ (i.e., $D = 1$ if either $S = 1$ or $K = 1$: Batman dies if he has a heart attack or the canonball kills him); $S = H$ (Batman's heart stops if he has a heart attack); and $K = \neg S \wedge C$ (Batman is killed by the canonball if the canonball hits him and his heart is still beating). We now get that Batman's heart attack rather than its absence is a cause of him being alive rather than dead. Clearly $(M, \vec{u}) \models H = 1 \wedge D = 1$. If we fix $K = 0$ (its actual value, since the cannonball in fact does not kill Batman; he is already dead by the time the cannonball hits him), then we have that $(M, \vec{u}) \models [H = 0, K = 0](D = 0)$, so AC2 holds. Thus, the causal part of H2 holds. (See [13, Example 2.3.3] for a detailed discussion of an isomorphic example.)

If we further assume, quite reasonably, that Batman prefers being alive to being dead (so the utility of being alive is higher than that of being dead) and that the default utility is that of him being alive, then H1 and H2 hold. Thus, our definition of harm avoids the counterintuitive conclusion by observing that Batman's heart attack caused his death, thereby causing the utility to be lower than the default. □

Our analysis of preemption is indicative of the more general point that many of the issues plaguing the literature on harm can be resolved by making use of causal models and the definitions of causation that they allow. Causal models allow a more precise and explicit representation of the relevant causal structure, thereby forcing a modeler to make modeling choices that resolve the inherent ambiguity that comes with an informal and underspecified causal scenario. Obviously such modeling choices can be the subject of debate (see [14] for a discussion of these modeling choices). The point is not that using causal models by itself determines a unique verdict on whether harm has occurred, but rather that such a debate *cannot even be had* without being explicit about the underlying causal structure.

## 4.2 Failing to Benefit

One of the central challenges in defining harm is to distinguish it from merely failing to benefit. Although most authors define benefit simply as the symmetric counterpart to harm, we do not believe that this is always appropriate; we return to this issue in [2] where we consider more quantitative notions of harm. But for the current discussion, we can set this issue aside: what matters is that merely failing to make someone better off does not in itself suffice to say that there was harm. Carlson et al. [5] present the following well-known scenario to illustrate the point.

**Example 3 (Golf Clubs)** Batman contemplates giving a set of golf clubs to Robin, but eventually decides to keep them. If he had not decided to keep them, he would have given the clubs to Robin, which would have made Robin better off.

By keeping the golf clubs, Batman clearly failed to make Robin better off. The counterfactual account considers any such failure to result in harm. Indeed, it is easy to see that C1–C3 hold. If we take $GGC$ to represent whether Batman gives the golf clubs to Robin ($GGC = 1$ if he does; $GGC = 0$ if he doesn't) and the outcome $O$ to represent whether Robin gets the golf clubs ($O = 1$ if he does; $O = 0$ if he doesn't), then $GGC = 0$ is a but-for cause of $GGC = 0$, so C1 and C2 hold. If we further assume that Robin's utility of getting the golf clubs is higher than his utility of not getting them, then C3 holds. Yet it sounds counterintuitive to claim that Batman harmed Robin on this occasion. □

Although H2 holds in our account of harm (for the same reason that C1–C3 hold), we avoid the counterintuitive conclusion by assuming that the default utility is $\mathbf{u}(O = 0)$, so H1 does not hold. This seems to us reasonable; there is nothing in the story that suggests that Robin is entitled to expect golf clubs. On the other hand, if we learn that Batman is a professional golfer, Robin has been his reliable caddy for many years, and that at the start of every past season Batman has purchased a set of golf clubs for Robin, then it sounds quite plausible that the default is for Robin to receive a set of golf clubs. With this default, H1 does hold, and our definition concludes that Robin *has* been harmed. Thus our account can offer different verdicts depending on the choice of default utility. As we said in the introduction, we view this flexibility as a feature of our account. This point is highlighted in the following, arguably more realistic, scenario. (RBT make exactly the same point as we do when they analyze such examples [22, p. 15].)

**Example 4 (Tip)** Batman contemplates giving a tip to his waiter, but eventually decides to keep the extra money for himself. If he had not decided to keep it, he would have given it to the waiter, which would have made the waiter better off.

To those living in the US, it does not at all sound counterintuitive to claim that Batman harmed the waiter, for his income substantially depends on receiving tips and he almost always does receive a tip. Indeed, if we take the default utility to be that of receiving a tip, then in this example, the waiter is harmed by Batman not giving a tip. By way of contrast, in countries in Europe where a tip would not be expected, it seems to us reasonable to take the default utility to be that of not receiving a tip. In this case, the waiter would not be harmed. □

Examples 3 and 4 are isomorphic as far as the causal structure goes; we can take the utilities to be the same as well. This means that we need additional structure to be able to claim that the agent is harmed in one case and not the other. That additional structure in our framework, which we would argue is quite natural, is the choice of default utility. Note that neither scenario explicitly mentions what the default utility should be. We thus need to rely on further background information to make a case for a particular choice. There can be many factors that go into determining a good default. We therefore do not give a general recipe for doing so. Indeed, as we pointed out in the introduction with the euthanasia example, reasonable people can disagree about the appropriate default (and thus reach different conclusions regarding harm).

## 4.3 Preventing Worse

There exist situations in which the actual event rather than an alternative event causes a bad outcome rather than a good outcome, but the alternative results in an even worse outcome. Because of the latter, we do not consider these situations to be cases of strict harm, due to condition H3 in Definition 2. From the perspective of the car manufacturer, this is precisely what is going on in our starting Example 1, but Bob might disagree. We now take a closer look at this example to bring out the conflicting perspectives.

**Example 5 (Autonomous Car)** Let $O$ be a three-valued variable capturing the outcome for Bob, with the utility defined as equal to the value of $O$. $O = 0.5$ stands for the injury resulting from crashing into the safety fence, and a potentially more severe injury resulting from crashing into the stationary car is captured by $O = 0$. Bob not being injured is $O = 1$.

Recall that the system has the built-in standard that the driver's reaction time is 10 seconds, which is too long to avoid colliding into the stationary car. Imagine the manufacturer implemented this standard by restricting the system's actions in such cases to two possibilities: do not intervene ($F = 0$) or drive into the fence ($F = 1$). This means that the causal structure is very similar to our Late Preemption example (Example 2), for hitting the fence preempts the collision with the stationary car. We therefore add a variable to capture the asymmetry between hitting the fence and hitting the stationary car: $FH$ and $CH$ respectively. The equation for $O$ is then such that $O = 1$ if $FH = CH = 0$, $O = 0.5$ if $FH = 1$, and $O = 0$ if $CH = 1$ and $FH = 0$.

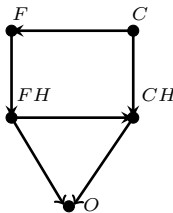

Figure 1: Causal graph for Ex. 5

As the autonomous car drives towards the fence only because there is a stationary car, the equation for $F$ is $F = C$ (where $C$ represents the presence of the car). The fact that hitting the fence prevents hitting the car is captured in the equation for $CH$: $CH = C \wedge \neg FH$. Lastly, we have $FH = F$. The context is such that $F = 1$ and $C = 1$, and thus $FH = 1$, $O = 0.5$, and $CH = 0$.

Did the system harm Bob? Carlson et al. [5] answer this in the negative for an example that is essentially the same as this one (see their "Many Threats" example), and use this verdict to argue against the causal-counterfactual account, which reaches the opposite verdict. According to them, the contrastive causal-counterfactual account does not reach this verdict because there is no contrastive causation here, but as they do not give a definition of causation, it is impossible to reconstruct how they arrive at this verdict. In any case, what matters for our purposes is that our definition of causation does consider the system's hitting the fence rather than not intervening to cause Bob being mildly injured rather than not being injured at all. To see why, observe that taking $\vec{W}$ to be $CH$, we get that $F = 1$ rather than $F = 0$ causes $O = 0.5$ rather than $O = 1$: $(M, \vec{u}) \models [F \leftarrow 0, CH \leftarrow 0]O = 1$.

Therefore, if we assume that the default utility is that of Bob not being injured, conditions H1 and H2 are satisfied and there is harm. Notice though that $F = 1$ rather than $F = 0$ is also a but-for cause of $O = 0.5$ rather than $O = 0$, that is, Bob's being mildly injured rather than severely injured counterfactually depends on the system's action. This is where condition H3 kicks in: it ensures that we do not consider there to be strict harm caused if the alternative would have resulted in an even worse outcome. Thus, the car manufacturer could make the case that although their policy harmed Bob, it was justified in doing so.

More generally, it is an easy consequence of our definitions that in cases where $\vec{X} = \vec{x}$ rather than $\vec{X} = \vec{x}'$ causes harm but not strict harm, the alternative event *would* have resulted in strict harm, i.e., $\vec{X} = \vec{x}'$ rather than $\vec{X} = \vec{x}$ would have caused strict harm. As a result, it is sensible in such cases for someone to argue that they were justified in causing harm, as the alternative would have been worse.

Bob, on the other hand, believes he has been strictly harmed, because he claims that he could have prevented the collision if he had been alerted. This disagreement can be captured formally by stating that Bob is using a three-valued variable $F$ instead of a binary one, where the third option ($F = 2$) corresponds to alerting Bob. Incorporating this variable into the model (and assuming that Bob is correct regarding his driving skills) we would again get that $F = 1$ rather than $F = 2$ causes $O = 0.5$ rather than $O = 1$, but with the important distinction that H3 *is* satisfied for these contrast values and thus the system's action does strictly harm Bob. Our analysis does not resolve the conflict (and it is not meant to do so), instead it allows for a precise formulation of the source of the disagreement. □

In the supplementary material, we describe four more examples illustrating multiple contrasts for the outcome, the role of the choice and the range of variables and the choice of default, and our rationale for considering a contrastive definition, rather than causal-counterfactual one.

## 5 Comparison to RBT

In this section, we do a more careful comparison of our approach and that of RBT. RBT focus on choices made by an agent, where these are choices of what action to take, and assume that there is a default action, to which they compare the choice made by the agent. It follows easily from RBT's definition that if the agent performs the default action, there is no harm. Yet there are many instances in which doing what is morally preferable causes harm, albeit accidentally. Simply imagine a doctor prescribing medication to a patient, and the patient unfortunately suffering a very rare allergic reaction to the medication, where the reaction is far worse than the initial condition that the patient had. Then clearly the doctor harmed the patient. The most obvious choice of default action here is the actual action (and, in fact, RBT themselves mention following "clinical guidelines" as an example of a default action in their Appendix C). But this means that according to RBT's definition there would not be harm here. Although we use a default utility, there is no need for this default utility to be the utility of a default action, so we do not have this problem.

Another significant difference between our approach and that of RBT is that, although RBT use causal models, unlike us, they do not use a sophisticated definition of actual causality such as the one given by Halpern [12, 13]. In their Definition 3, RBT consider but-for causality. Not surprisingly, this will not suffice to deal with problematic examples where the more general notion of causality is needed (see, e.g., Example 2). In their Definition 9, they generalize Definition 3 to allow for path-dependent causality. They do not explain how they choose which paths to consider, but in both Example 2 and the corresponding example of late preemption they consider, by choosing the appropriate paths, they can simulate the effects of AC2. (Specifically, they can simulate the effect of choosing a set $\vec{W}$ of variables and fixing them to their actual values.) As a consequence, they get the same results as those obtained by Halpern's definition of actual causality. It is not clear whether this will always be the case. More importantly, the ability to determine harm relative to some choice of paths gives the modeler a significant extra degree of freedom to tailor the results obtained. We believe that if paths are going to be used, there needs to be a more principled analysis of how to go about choosing them.

## 6 Conclusion

We have defined a qualitative notion of harm, and shown that it deals well with the many problematic examples in the philosophy literature. We believe that our definition will be widely applicable in the regulatory frameworks that we expect to be designed soon in order to deal with autonomous systems.

Of course, having a qualitative notion of harm is only a first step. For practical applications, we often need to quantify harm; for example, we may want to choose the least harmful of a set of possible interventions. As we said, we develop a quantitative notion of harm in [2]. While one could just define a quantitive notion that considers the difference between the utility of the actual outcome and the default utility (this is essentially what RBT do), we believe that the actual problem is more nuanced. For example, even if we can agree on the degree of harm to an individual, if there are many people involved and there is a probability of each one being harmed, should we just sum the individual harms, weighted by the probability? We argue that this is not always appropriate, and discuss alternatives, drawing on work from the decision-theory literature.

## Acknowledgements

The authors would like to thank the NeurIPS reviewers for their detailed comments and Jonathan Richens for a fruitful discussion of a preliminary version of this paper. Sander Beckers was supported by the German Research Foundation (DFG) under Germany's Excellence Strategy – EXC number 2064/1 – Project number 390727645, and by the Alexander von Humboldt Foundation. Hana Chockler was supported in part by the UKRI Trust-worthy Autonomous Systems Hub (EP/V00784X/1) and the UKRI Strategic Priorities Fund to the UKRI Research Node on Trustworthy Autonomous Systems Governance and Regulation (EP/V026607/1). Joe Halpern was supported in part by NSF grant IIS-1703846, ARO grant W911NF-22-1-0061, and MURI grant W911NF-19-1-0217.

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
