## A Appendix

Here we provide some additional details on three topics. First, we illustrate the role played by H3 in our definition of harm. Second, we discuss in more detail how our approach differs from that of RBT. Third, we present four more examples that illustrate how our definition handles issues which have been discussed in the harm literature.

### A.1 Discussion of H3

As we mentioned, H3 is intended to capture the intuition that preventing a worse outcome is not harmful. For example, following the reasoning of the car manufacturer in Example 5, the system's decision to drive into the fence rather than doing nothing is not harmful because Bob would have suffered even worse injuries had the system done nothing. Since H1 and H2 are satisfied for this particular contrastive event, our definition would reach the opposite verdict if it weren't for H3. Note that the counterfactual comparative account (Definition 3) also says that there is no harm: the alternative event under consideration would have given a worse outcome, so that C3 is not satisfied, and therefore there is no harm. Considering H3 gives more insight into the differences between the counterfactual account and ours.

Suppose that we consider some contrastive event $\vec{X} = \vec{x}'$ such that $(M, \vec{u}) \models \vec{X} = \vec{x} \wedge O = o$ and $(M, \vec{u}) \models [\vec{X} \leftarrow \vec{x}'](O = o'')$, so C1 and C2 hold, and the first half of H2 holds if $o'' \neq o$: $\vec{X} = \vec{x}$ rather than $\vec{X} = \vec{x}'$ causes $O = o$ rather than $O = o''$. H3 plays no role if H1 is not satisfied, so for simplicity, suppose that H1 also holds. Then it is easy to see that whenever $\mathbf{u}(O = o) \neq \mathbf{u}(O = o'')$, our definition gives the same verdict as the counterfactual comparative definition for this particular contrast (i.e., for this choice of $\vec{x}'$): if $\mathbf{u}(O = o) < \mathbf{u}(O = o'')$, then $o'' \neq o$, so H2 holds, as do H3 and C3; it follows that both definitions declare $\vec{X} = \vec{x}$ a harm. On the other hand, if $\mathbf{u}(O = o) > \mathbf{u}(O = o'')$, then neither C3 nor H3 hold (for this choice of $\vec{x}'$).

What happens if $\mathbf{u}(O = o) = \mathbf{u}(O = o'')$? This can happen for two reasons:

1. there is no but-for causation, that is, $o = o''$;

2. there is but-for causation but the counterfactual outcome $O = o''$ happens to have utility identical to the actual outcome.

Thus, roughly speaking (and ignoring the key role played by the default utility), our definition differs from the counterfactual comparative account only if $\vec{X} = \vec{x}$ rather than $\vec{X} = \vec{x}''$ is not a but-for cause of the actual utility: changing $\vec{x}$ into $\vec{x}''$ does not change the agent's utility.

Examples in which the first reason is relevant are widespread and crucial to our analysis, for those are precisely the examples in which actual causation (Definition 1) and but-for causation come apart. Our Late Preemption example (Example 2) offers one illustration, the literature on actual causation contains many more. An example where the second reason is relevant involves a more subtle way in which but-for causation comes apart from actual causation. Consider a "Sophie's choice" like setting: An agent must choose whether $X = 1$ or $X = 2$. There are two children, who will either live or die depending on the choice: if $X = i$ is chosen, then child $i$ lives ($L_i = 1$) and child $3-i$ dies ($L_{3-i} = 0$). The possible outcomes are that both children live ($o_{11}$), just child 1 lives ($o_{10}$), just child 2 lives ($o_{01}$), and neither child lives ($o_{00}$), where $d = \mathbf{u}(O = o_{11}) > \mathbf{u}(O = o_{10}) = \mathbf{u}(O = o_{01}) > \mathbf{u}(O = o_{00})$. In fact, $X = 1$ is chosen, so we get but-for causality, but switching from $X = 1$ to $X = 2$ gives an outcome of equal utility. However, if we hold $L_1 = 1$ fixed (which we can do in our framework to show causality) and switch to $X = 2$, then we get the outcome $O = o_{11}$. Thus, in our framework $X = 1$ harms the agent; in the causal counterfactual framework, it does not.

This emphasizes the point we (and RBT) made that one set of problems that occur in defining harm is identical to the type of problems that occur in defining causation, and can be solved in the same way.

### A.2 Comparison to RBT

In this section, we do a more careful comparison of our approach and RBT's approach. RBT restrict their analysis to choices made by agents, where different choices can be taken to have different normative content (i.e., some choices are more normatively appropriate than others, although people

might disagree as to which is the more appropriate choice, as in our euthanasia example). This assumption is critical for them, since it plays a key role in how they determine both the default action and the contingency to hold fixed when checking condition AC2 in the definition of causation (Definition 1). There are several problems with this approach.

First, as has often been pointed out in the harm literature (and is critical to the insurance industry!) harm can be caused by events other than agent's actions. Indeed we already came across such an example: Batman getting a heart attack in Example 2 causes Batman harm. We would certainly like an account of harm that applies to such "natural events".

Second, by construction of their definition, whenever an agent performs the default action, there is no harm according to RBT. Yet there are many instances in which doing what is morally preferable causes harm, albeit accidentally. Simply imagine a doctor prescribing medication to a patient, and the patient unfortunately suffering a very rare allergic reaction to the medication, where the reaction is far worse than the initial condition that the patient had. Then clearly the doctor harmed the patient. The most obvious choice of default action here is the actual action (and, in fact, RBT themselves mention following "clinical guidelines" as an example of a default action in Appendix B). But this means that according to RBT's definition there would not be harm here.

Third, even if we focus on choices made by agents and assume that there's a sensible default action, there is a key difference between our definition of causality, which, as we said, is based on that of Halpern [11, 12], and that of RBT (given in their Appendix A as Definition 9). Whereas in AC2 we existentially quantify over the set $\vec{W}$, RBT give a definition of harm relative to a fixed set $\vec{W}$, and assume that $\vec{W}$ should be determined by normative considerations (as they say at the end of their Appendix B, "when establishing harm the conditional contingency [i.e., choice of $\vec{W}$] corresponds to a single contingency that is determined a priori based on our normative assumptions, and taking the wrong contingency (or allowing for any contingency) will result in harm or benefit being misattributed"). Nonetheless, RBT claim that their approach is equivalent to that of Halpern, which is clearly not the case.

On a conceptual level, the same points arise for a normatively determined contingency as the ones we brought up for the default action: we would also like to apply the notion of harm to natural events (and thus to cases in which there does not seem to be any contingency that is morally preferable over others), and there can be situations in which doing the right thing causes harm. Perhaps RBT could try and resolve this by allowing there to be multiple contingencies that can be used when applying Definition 9, but then they would have to somehow aggregate the different harms that we get for each specific contingency; it is not at all clear how this would be done.

To defend their use of default actions and the idea of having a normatively determined contingency, RBT consider two examples. In the first, Bob expects a government check for $100, but does not get one because, instead, Alice puts $100 into his bank account, which disqualifies him from government assistance. In the second, there are two do-gooders, Alice and Eve, who conspire to lift Bob out of poverty. Alice gets there first, giving Bob $100, but if she had not done so, Eve would have. We agree with RBT that, in both examples, Alice is the cause of Bob getting $100 rather than 0 (and this follows easily from our definitions). We also agree that in the first case, Alice's action does not benefit Bob, while in the second it does. Although we do not give a definition of benefit in our paper, taking the obvious analogue of our definition of harm, we would get this result by simply taking different defaults in the two examples: the default in the first is that Bob gets $100 (because that is the societal expectation, given the government program) while in the second it is that Bob gets $0. We still get the arguably "right" answer although we existentially quantify in AC2. Using the contingency only to establish causality as we do (rather than as a way to establish the amount of harm, as RBT seem to do), we can still deal with all the examples, while also being able to deal with cases where there are no obvious normative considerations that determine the appropriate contingency.

## A.3   More Examples

We present four further examples that illustrate how our approach deals with the difficulties of defining harm that have been highlighted in the literature.

The cases in Sections 4.1 and 4.2 all involved a binary outcome; there were only two relevant events that could occur. Carlson et al. [4] discuss cases that involve more than two possible events in order to argue against existing causal accounts. The following example forms one instance of their argument.

**Example 6 (Tear Gas)** The Joker sprays tear gas in exactly one of Batman's eyes. If he had not done that, he would have sprayed tear gas in both of Batman's eyes, which would have made Batman even worse off. One of the alternatives available to the Joker, however, was to simply leave Batman alone.

Intuitively here Joker harms Batman when he sprays him. To argue that the "incorrect" answer is obtained by the definition of harm they focus on, Carlson et al. consider a specific alternative event, namely, that Joker sprays tear gas in both of Batman's eyes, while observing that other alternatives (like leaving Batman alone) are also available. Rather than existentially quantifying over $\vec{x}'$, as we have done, (both in Definition 2 and the gloss of the counterfactual harm definition given in Definition 3), they take a version of counterfactual harm where $\vec{X} = \vec{x}'$ is taken to be the closest alternative to $\vec{X} = \vec{x}$ (according to some implicit, but unspecified, notion of closeness). Both our definition of harm and our gloss of the counterfactual definition (with the obvious assumptions about utility, and taking the default utility to be that of Batman being unharmed for our definition) agree that Joker did harm Batman in this case, as we would expect.

In this example, there are three events of interest (Joker sprays tear gas in one eye; Joker sprays tear gas in both eyes; Joker doesn't spray tear gas at all). We can model this using a variable $TG$ that takes on three possible values (say, 0, 1, and 2). According to Definition 3, as long as one of them leads to a better utility than what actually happened, there was harm. But as the golf clubs example shows, this conclusion is not always justified; in general, we need to take defaults into account. □

Now we present an example, due to Shiffrin [25], that illustrates the role of both the choice of the range of variables in the causal model and the choice of default.

**Example 7** Betty is drowning in a fast-moving river. Veronica rescues her by grabbing her arm and pulling her out, accidentally fracturing Betty's humerus.

Did Veronica's rescue harm Betty? Shiffrin claims it does because Veronica could have pulled her out without breaking her arm. Indeed, Klocksiem [18], in his analysis, points out that "it seems possible to rescue someone from drowning without breaking her arm". The first step in our analysis is to decide whether we should allow this possibility. That is, suppose that we have a variable $P$ that describes how and whether Veronica pulls out Betty. We can take $P = 0$ if Veronica does not pull out Betty, $P = 1$ if she pulls her out by grabbing (and breaking) her arm. The modeler must then decide whether to allow $P$ to take a value, say 2, where $P = 2$ if Veronica rescues Betty in such a way that Betty's arm is not broken. Reasonable people might disagree whether such an event is possible. First suppose we decide that $P$ can take only values 0 and 1. Then the possible outcomes are that Betty drowns ($O = 0$) or Betty is saved ($O = 1$). In this model, any utility function that makes the utility of drowning worse than that of being saved would result in Veronica's rescue not harming Betty.

Now suppose that we allow $P = 2$. Then we would take $O = 1$ to represent Betty being saved but her arm being broken, and $O = 2$ to represent Betty being saved without her arm being broken. In that case, whether Veronica harms Betty depends on the default. If we take the default utility to be $\mathbf{u}(O = 2)$ then Veronica does cause Betty harm, while if we take it to be $\mathbf{u}(O = 0)$, she does not. Note that the latter choice is quite defensible. Given Betty's situation, making it out alive in whatever way possible would presumably be all that matters to her. □

This example clearly shows that to apply our framework in practice, it is important to have some guidelines on what count as a reasonable choice, both in the choice of variables and values and the choice of default value. As we mentioned in the introduction, Halpern and Hitchock [13] discuss this issue in the context of causal models; to the best of our knowledge, this issue has not been discussed in the context of default values. While this issue is beyond the scope of the current paper, we should make clear that we would not, in general, expect there to be a unique "correct" model. As we have said repeatedly, reasonable people can disagree about these choices.

There is one final issue we would like to address: why we consider a contrastive definition rather than just giving a definition in the spirit of the causal-counterfactual account. Definition 2 explicitly invokes a contrastive outcome $o'$ whose utility is better than that of the actual outcome. We could

have instead just defined harm as the result of causing an outcome whose utility is worse than the default.

One reason why we did not do so is that the default utility is not always achievable, and it would be counterintuitive to say that the agent was harmed if the outcome has a utility lower than the default, even though it is the best possible outcome. For example, there are diseases for which a surgery can only provide a temporary cure; in this case, a successful surgery gives the patient a temporary relief, and an unsuccessful surgery results in the patient's death. While the default utility for the patient, as for all people, is to be alive and healthy, saying that a successful surgery harmed the patient seems wrong. In fact, defining harm as the result of causing an outcome with the utility worse than the default provides counter-intuitive results even when the default utility is achievable, as the following example demonstrates.

**Example 8 (Pills)** Consider the following vignette, again taken from [4] (where it is presented as a problem for both the causal-counterfactual and contrastive causal-counterfactual accounts):

> Barney suffers from a painful disease. On Monday, he can either take Pill A or not. On Tuesday, he will have another choice, between taking Pill B or not. Barney believes that he will be completely cured just in case he takes only Pill A, and partially cured just in case he takes both pills. Accordingly, he takes Pill A on Monday and does not take Pill B on Tuesday . . . He is, however, misinformed about the effects of the pills. Taking only Pill A causes his disease to be merely partially cured. If he had taken both pills, he would have been completely cured. Had he not taken Pill A on Monday, on the other hand, nothing he could have done later would have produced even a partial cure.

To capture this in our framework, let $O$ be a three-valued variable that captures Barney's health: $O = 2$ if he is fully cured, $O = 1$ if he is partially cured, and $O = 0$ if he is not cured at all. $A$ and $B$ capture whether or not Barney takes pills A and B respectively. The equation for $O$ is then such that $O = 2$ if $A = B = 1$, $O = 1$ if $A = 1$ and $B = 0$, and $O = 0$ otherwise. As Barney considers taking pill B only if he fails to take pill A, the equation for $B$ is $B = \neg A$. The context is such that $A = 1$; therefore, $B = 0$ and $O = 1$.

Carlson et al. claim that taking the pill does not harm Barney; we agree. Yet it easy to see that $A = 1$ does cause $O = 1$. Indeed it is a but-for cause: had Barney not taken the pill, $O$ would have been $0$. It is easy to see why this is a problem for the causal-counterfactual account: Barney would have been better off if $O = 1$ had not obtained; specifically, he would be better off if $O$ had been $2$ (although this is not the outcome that results when changing $A$ to $0$ and therefore is not a problem for the counterfactual comparative account). Carlson et al. also view it as a problem for the contrastive causal-counterfactual account, because in applying it, they compare $O = 1$ to the outcome $O = 2$, (which, again, is not the outcome that obtains by switching $A$ to $0$), since they take the closest world to the one where Barney takes just one pill to be the world where he takes both pills. Our definition avoids this problem. We do not consider the "closest" state of affairs. Rather, we compare $O = 1$ to the outcome $O = 0$ caused by switching to $A = 0$. $O = 0$ has utility worse than that of the outcome obtained from $A = 1$, so it is not a harm according to our definition, for what we view as the "right" reasons. Assuming that the default utility is $\mathbf{u}(O = 2)$, $A = 1$ does cause an outcome whose utility is worse than the default and therefore a non-contrastive version of our definition would not have given the desired result. □