# OpenReview forum: "A Causal Analysis of Harm"
_NeurIPS.cc/2022/Conference — NeurIPS 2022 Accept_

### Official Review · Reviewer_RbEE · 2022-07-15

**Rating:** 7
**Confidence:** 2
**Soundness:** 3 good
**Presentation:** 3 good
**Contribution:** 3 good

**Summary:**

The work proposes a qualitative definition of harm based on causal models in the context of dealing with autonomous AI systems. The authors state that the notion of harm is ill-defined and motivate their qualitative approach on the observation that much of the discussion of harm is qualitative. After reviewing causal models and describing the introduced definition of harm, the work applies and discusses it (including its limitations) on various examples.

**Questions:**

See W1 and W2 above.

**Ethics Review Area:**

["I don’t know"]

**Limitations:**

The authors adequately addressed the limitations.

**Strengths And Weaknesses:**

**Strengths**

- S1: The definition is inspired by previous work (RBT, Bontly) and extends them to a causal definition.

- S2: Besides some minor issues (see following weaknesses), the proposed definition is well described and understandable.

- S3: The extensive examples, including the discussions contained in the main text and the supplement, give a great overview and intuition of the proposed definition.

**Weaknesses**
- W1: While building on Y. Halpern [11] and Y. Halpern and J. Pearl. [13] working with the concept of 'actual causality', the paper might be improved by briefly discussing differences to the common approach of probabilistic structural causal models [X1, X2], where variables are considered to be sampled from probability distributions with structural equations including random noise terms. Especially recursive deterministic computation of variables as described in the paper must assume that no noise terms are included in the system. The authors might consider to state this assumption explicitly.

- W2: Furthermore, the relations between exogenous and endogenous variables (line 112) seems to be more strict than the standard definition using 'Pearlian causality'. E.g. compare to [X2] p.23 “Endogeneous variables are those that the modeler tries to understand, while exogenous ones are determined by factors outside the model, and are taken as given.” While similar in the intention, Peters et al. give no definition of how to treat the relationship between endogenous and exogenous variables. Assuming endogenous variables to be fully determined by exogenous factors is a strong assumption that, again, only holds under the assumption that all exogenous variables are known and observed and no additional random noise terms are influencing the system.


[X1] Pearl, J. (2009). Causality (Cambridge university press)

[X2] Peters, J., Janzing, D., and Schölkopf, B. (2017). Elements of causal inference: foundations and learning algorithms (The MIT Press)


**Minor comments**

- Typo line 104 "that that"
- Typo line 166 "beem"

---

> ### Author Response · Authors · 2022-08-01
> **Reply to reviewer**
>
> W1: We will add an explanation that distinguishes our framework from
> the probabilistic one and state the assumption of the absence of
> noise.
>
> W2: Indeed, our definitions are with respect to a given causal model,
> assuming that all relevant variables and their structural equations
> are known and included in the model. This is a standard assumption in
> actual causality.
> But note that once we move to quantitative harm, we do allow there to
> be a probability distribution on contexts, so we are much closer to the
> setting of Peters et al.

---

> > ### Comment · Reviewer_RbEE · 2022-08-07
> > **RE: Reply to reviewer**
> >
> > Thank you for your comment and the clarification.
> >
> > I also keep my previous rating.

---

### Official Review · Reviewer_7g7A · 2022-07-16

**Rating:** 6
**Confidence:** 4
**Soundness:** 2 fair
**Presentation:** 4 excellent
**Contribution:** 3 good

**Summary:**

This paper proposes a way to define **harm** in a causal way. The work in more on of a philosophical level, trying to criticize the fact that the definition of harm in philosophical studies is generally devoid of causality and putting forward the notion that the use of causality theory can define a more general definition of harm. Several examples are presented throughout the paper that show how the causal analysis and a new definition of harm can handle several standard examples from literature and can thus be used for effective reasoning in various domains and situations.

**Questions:**

In addition to some important points raised in the weaknesses section, I have a couple of specific questions:

1. How will the definition of harm be extended/modified if we have dependent exogeneous variables in the causal graph?

2. How are confounders handled in the causal definition of harm?

**Limitations:**

This is tricky. Although I do not see any specific negative societal impact, defining and constraining harm to a single definition might be a little bit risky. As I have mentioned in my review before, each can have his/her own interpretation with respect to the definition.

**Strengths And Weaknesses:**

Strengths:

1. The paper tackles a very important and sensitive problem of defining harm. Since this can have multiple interpretations and can be conceived by different people in unique ways, trying to  come up with a causal-standard definition is indeed a nice effort.

2. The paper is written beautifully and the examples are crystal clear. In my opinion, even a person with little knowledge of causality will be able to follow the paper and get an overall idea about the main message of this work.

3. The definition of **harm** in section 3 seems well thought out and well sketched.

Weaknesses:

1. The method seems to have a very heavy dependence on the default utility.

2. I do not agree with the Batman and Robin example, where the definition concludes that Robin was harmed. Just because Robin receives a new set of clubs at start of every season, it still does not make him entitled to receive it the next year as well. There can be the effects of some hidden confounders that are being ignored here.

3. The default utility of receiving or not receiving a tip in **Example 4** seems incomplete. For example, what about the cases where the exact amount of tip is also important? Say the waiter is better off if he/she receives 15% of the total bill as a tip instead of 5% in the US.

Overall, this is an interesting take combining philosophical arguments with causality in a more explicit way. I would be eager to see the author's response to the raised points.

---

> ### Author Response · Authors · 2022-08-01
> **Reply to reviewer**
>
> Reply to weaknesses:
>
> 1. Indeed, the decision of whether harm was caused relies on the
> causal model, and in particular, the utility function. In fact, all
> concepts related to actual causality (cause, blame, responsibility,
> etc.) are relative to a causal model. The default value of
> the utility function is also a part of the causal model.
>
> 2. We did not examine the possibility of having hidden confounders at
> all. We assume that the model is known and is given to
> us in its entirety. In other words, if something is not in the model, then
> it is not relevant. Your point about Batman and Robin is a good
> illustration of the fact that everything is relative to the model.
> If we had a model that explicitly described all the confounders in the
> Batman-Robin example, we might well get different results.
> See also the discussion on the autonomous car example - having different
> views about the default utility results in different decisions
> regarding harm caused to Bob.
>
> 3. Indeed, the exact amount of tip is very important. In the current
> qualitative definition, we decide only whether harm occurred. The
> quantitative definition, measuring the amount of harm, is the subject
> of our current work. By the way, in the waiter example, one could also
> argue that receiving any tip is a benefit, so a waiter is not harmed
> if they receive a tip. We could capture this by simply taking a
> different default.
>
> Reply to questions:
>
> 1. We are not sure what you mean by dependent exogenous variables. In
> structural causal models,
> dependencies between variables are characterized by equations.  But
> for exogenous variables, there are no equations.  Rather, we just have
> a context that determines their values.  So, at least with the obvious
> interpretation, the notion of dependent exogenous variables doesn't
> make sense in the deterministic causal models we work with.  (Of
> course, if we add probability in the standard way, by putting a
> probability on exogenous variables, there would be probabilistic
> dependencies between variables.  But this would not cause any problems
> for our definition.)
>
> 2. Confounders, if they are defined explicitly in the causal model, are
> like other variables; their interaction with other variables is
> described by the structural equations.
> As we said, we (implicitly) assume that the model given to us as
> an input contains all the relevant variables.

---

> > ### Comment · Reviewer_7g7A · 2022-08-07
> > **Thank you for the response**
> >
> > I keep my rating as is.

---

### Official Review · Reviewer_f27v · 2022-07-17

**Rating:** 5
**Confidence:** 3
**Ethics Flag:** Yes
**Soundness:** 2 fair
**Presentation:** 4 excellent
**Contribution:** 3 good

**Summary:**

In this paper, the authors proposed the first qualitative causal framework that is used to define the harm and conceptually show their framework can solve various controversial real-life examples in terms of minimizing the harm.

**Questions:**

The authors applied the framework of harm to several real-life scenarios. However, one thing in common is that all the scenarios consider defining harm for a single agent. However, in reality, the harm is usually multi-agents based. Without a clear quantitative suggestion for the utility function, it is hard to minimize the total harm. Even without considering the dependency among the agents, simply using the framework proposed by the author to make the decision could be challenging (Consider the Trolley problem). I suggest the authors consider how to address this issue or explicitly state this limitation.


**Ethics Review Area:**

["Discrimination / Bias / Fairness Concerns"]

**Limitations:**

The utility function u which measures the goodness of the decision is a vague term. In reality, the way of defining such a utility function could invoke controversy and fairness issues, which have also been discussed by the author.

**Strengths And Weaknesses:**

Strength:
It is by far the first literature, as far as I know, that attempts to discuss forming the framework for evaluating harm. It also discussed how to apply this framework to several real-life problems.

Weakness:

Though I understand the difficulty of quantitatively defining the utility function, the lack of discussion of it could discourage its application to reality, I will discuss it in detail in the questions session.

---

> ### Author Response · Authors · 2022-08-01
> **Reply to reviewer**
>
> We agree that real situations are more complex than harm to a single
> agent in a deterministic scenario. We are writing the next paper,
> where we discuss aggregation of harm to multiple agents and also
> probabilistic settings. This indeed requires a quantitative measure of
> harm, which we also define in the next paper. As coming up with a
> qualitative notion of harm is already quite challenging, as you point
> out, the current paper focusses on the qualitative definition and does
> not discuss aggregation.
> Note that we mention the subtleties involved with aggregation in the
> conclusion.  We will make the fact that it is a limitation of this
> paper not to deal with it explicit.

---

> > ### Comment · Reviewer_f27v · 2022-08-07
> > **Response to author**
> >
> > Thanks for your reply, I keep my rating.

---

### Official Review · Reviewer_2wZ6 · 2022-07-21

**Rating:** 5
**Confidence:** 3
**Soundness:** 3 good
**Presentation:** 2 fair
**Contribution:** 2 fair

**Summary:**

[REMARK: This is an emergency review, that is, the reviewer did neither select autonomously into this paper out of interest/expertise nor was the reviewer granted with sufficient time (with respect to standard time frames) to appropriately provide a review. Please take the review with caution, the confidence level will be adjusted accordingly.]

The paper presents a formalization of the concept of "harm" (to harm someone) using structural causal models (SCM) as found in the counterfactual theory of causality vocally discussed by Judea Pearl. Specifically, the authors extend SCMs to cover a "default utility" (referred to as causal utility model) to then propose 3 conditions (H1 to H3) which captures (defines) whether an even $\overset{\rightarrow}{x}$ causes harm to an agent $\mathbf{ag}$. The authors provide a motivation, a comparison to other definitions (in terms of their presented formalism) and finally discuss several philosophical key examples displaying the advantages of their proposed definition.

**Questions:**

The questions are derived mostly from aspects mentioned in the "Weaknesses" section. I'd invite the authors to answer these questions to overcome the observed weaknesses for improving the paper to ultimately increase the chances of both contribution and visibility to the community. The list of questions is unordered and they range in quality and speed of expected answer (some are minor, quickly answerable whereas others are more fundamental, crucial, maybe difficult to answer):
* How can we cope with temporal notions?
* How can we overcome intractability of harm computation, in the face of our necessity to compute since otherwise a formalized notion would have no practical implications?
* Under which circumstances can our causal model guarantee that our harm computation will be approximately correct?

**Limitations:**

No contradictions or any sort of relevant mistake have been detected in the paper. The overall clarity of the paper is a clear advantage. Existing bodies of work are being referenced accordingly throughout the paper.

There is nothing to be reproduced, therefore, naturally, there is also no code. Societal impact is being discussed throughout.

**Strengths And Weaknesses:**

**Strengths**

Discussing and overall researching into harm and how to capture/measure it is certainly timely in the advent of discussions around AGI alignment etc. The same holds for the intention of deriving a causal account since causal models allow for an explicable, interpretable modelling scheme. As the authors suggest, one can only debate about assumptions if said modelling assumption are recognizable. The presented formalization copes well with the discussed examples, which cover key philosophical dilemmas. Capturing formally, with thought through notions, otherwise intangible concepts like harm is an important step towards accountable intelligent systems.

**Weaknesses**

While one can question on a more general note whether discussing this topic within NeurIPS (especially in the presented form) is sensible, I believe the importance on a societal scale for such a work is undeniable and NeurIPS with both its breadth and influence would fit as a great platform. To comment on the paper itself more importantly, the presentation sets its focus on the formal account of harm, and while it does motivate generally and discuss specifically, the presentation (of otherwise very visual/intuitive ideas) is arguably dry. That is, to convey the ideas or spread the formalized notion (which is likely the key interest of the authors) alternate means of presentation using illustrations with figures but also visualizations of possibly synthesized versions of the examples would help in doing so. Regarding the mathematical account, the lack of analysis for induced properties (when is the computation feasible, what models do satisfy the condition in the first place, etc.) is a weakness (albeit a likely difficult one to fix since it is difficult enough already to capture a philosophical core problem that has existed for centuries mathematically).

---

> ### Author Response · Authors · 2022-08-01
> **Reply to reviewer**
>
> We hope that the examples provided in the paper help to capture the intuition behind the proposed notion of harm. We will add
> visualisations as per your suggestion to the extent that we can fit them in, given space limitations. Regarding the computational
> complexity/feasibility, it is probably of little surprise that harm is intractable, since it is based on the definition of cause, which is DP-complete. However, since causal models tend to be quite small, in practice the computation is often efficient.
>
> 1. Temporal notions are expressed by the choice of variables in the causal model, and the relationships between them; see, e.g., the extended Suzy-Billy example in the original paper on actual causality ([HP01]). This is a standard approach in causal models.  We will comment on this in the paper.
>
> 2. See above.
>
> 3. This paper focuses on deciding harm, not computing it. Also, we analyse one context, where we know the values of the variables. Hence, determining whether harm occurred is precise. The question of approximation would be more interesting with a quantitative model of harm (which, as we say in the conclusion, we are currently working on), where we could try to compute an approximation to the actual degree of harm.

---

> > ### Comment · Reviewer_2wZ6 · 2022-08-06
> > **response to authors**
> >
> > Thank you for providing brief answers to the questions, however, could you be more specific?
> > For Q1, could you offer an example of how you would comment on this within the paper?
> > For Q2, I presume that "See above" refers to the first paragraph. However, the intention behind Q2 is to ask how we can justify a formal definition of harm in the first place (especially in the face of intractability).
> > For Q3, could you offer an example here as well?
> > Also, regarding the visualizations of the intuition using Figures, how would you go about achieving this?
> > Thank you.

---

> > > ### Author Response · Authors · 2022-08-07
> > > **Reply to reviewer**
> > >
> > > > For Q1, could you offer an example of how you would comment on this within the paper?
> > >
> > > As we consider this to be a technical issue having to do with causal models and defining causation itself, rather than an issue which is particular to defining harm, we would restrict ourselves to being explicit about the fact that the reference we provided (to [12]) is not just relevant for Late Preemption but handling temporal variables more generally.
> > >
> > > > For Q2, I presume that "See above" refers to the first paragraph. However, the intention behind Q2 is to ask how we can justify a formal definition of harm in the first place (especially in the face of intractability).
> > >
> > > Complexity results are worst case.  The fact that harm is intractable in the worst case does not make it useless!  We strongly suspect that, in many cases of interest, it will be quite tractable (an obvious case
> > > is when there aren't too many variables).  An interesting technical question would be to characterize cases of practical interest where the computation is tractable, although exploring this issue is beyond the scope of the current paper.
> > >
> > > > For Q3, could you offer an example here as well?
> > >
> > > As we said, this question is more relevant to the paper that we are currently writing on quantitative harm.  In the setting of this paper, where there is only one context and no uncertainty, computing quantitative harm is straightforward (it's essentially the difference between the utility that could have been attained and the max of the utility that was attained and the default utility).  So computing the quantitative harm is no harder than computing whether there was harm. But once we consider "societal harm", where we need to aggregate the harm caused to many people, approximation may play an important role.
> > >
> > > > Also, regarding the visualizations of the intuition using Figures, how would you go about achieving this? Thank you.
> > >
> > > Our plan was to add causal graphs (acyclic graphs that look like Bayesian networks) to illustrate the scenarios.  These diagrams should make it easier to illustrate some of the points regarding causality.

---

### Official Review · Reviewer_6SwV · 2022-07-21

**Rating:** 3
**Confidence:** 4
**Soundness:** 1 poor
**Presentation:** 3 good
**Contribution:** 2 fair

**Summary:**

The paper introduces a formal definition of harm that builds upon the modified Halpern-Pearl definition of actual causation. The main differences between the definition and previous definitions in the literature are the new concepts introduced in the *causal utility model*, which adds 1) a utility function $\mathbf u : \mathcal R(O) \to [0, 1]$ for some $O \in \mathcal V$ and 2) a "default utility" $d \in [0, 1]$ to the standard causal model. The authors introduce 7 examples of situations where an agent is or is not caused harm that are prominent in the literature. They argue that their definition of harm either correctly identifies the presence or absence of harm for each of these examples, or correctly identifies the precise source of ambiguity regarding the presence or absence of harm.

**Questions:**

Questions:
1. Why is the causal utility model defined and used, rather than Halpern's extended causal model in the definition of harm? It appears to me that harm as defined in the paper is equivalent to causation under an extended causal model with a particular partial preorder.
2. Is it possible for harm to occur without causation? If not, why is this? Is it possible to introduce a result demonstrating that harm should occur due to causation?
3. It seems the notion of harm and blame should be related; do you have thoughts about this? Shouldn’t the definition of harm incorporate the mental model of the agent taking an action that may have harmed the agent being harmed and that of the agent being harmed?
4. The authors emphasize that the definition of harm proposed in the paper is "qualitative", while RBT's definition is quantitative. I am not sure this is clear, can you explain this distinction and provide further elaborations on how to think about it?
5. If the definition of harm is qualitative, why does the causal utility model use a utility function that maps to a number in $[0, 1]$, rather than one that induces an ordering (or partial preorder) over $O$? As the authors note, it is unclear how to combine the numerical utilities of multiple agents. Is there a reason a mapping to $[0, 1]$ must be used rather than an ordering?
6. This definition of harm seems quite interesting, and I am a bit intrigued to know if counterexamples to the definition already exist (similar to the ones I showed in the limitations section).

**Limitations:**

I do not see potential for negative social impact of the work, as the work is aimed at formally defining harm - research which may help autonomous systems reason about minimizing harm in the future. Thus, I list some limitations of the work I observed below.

## Insufficient evaluation

The positive examples provided do not provide any guarantees that there are no counterexamples to the definition, and there are only 7 positive examples provided with no extensive human evaluation (by laypeople or by experts). I list two possible counterexamples below and would be curious to hear the authors’ thoughts.

## The relationship between harm and responsibility of agents is not considered by the work

This criticism is aimed at the fact that the definition proposed in the work does not consider the intention of the agents.

Consider the following example:

> A driver honks her car horn at the driver in front of her, startling an apartment resident dangling his legs off of his balcony railing. The resident falls off the balcony in fright, becoming gravely injured upon hitting the ground.

It seems strange to claim that the driver harmed the resident, because 1) she could not have known that the resident was in danger of falling 2) even if she did know, the resident is partially at fault for willfully placing himself in a precarious situation. To make the example even more clear-cut:

> An apartment resident decides that he will jump off his balcony upon the first car honk he hears. A driver honks her horn at the driver in front of her. Hearing this horn honk, the resident jumps off the balcony, becoming injured upon hitting the ground.

Now, it seems even stranger to claim that the driver harmed the resident, because the resident took an action intended to harm himself, while the driver did not.

This issue could be resolved by using Halpern's concept of blame introduced in (Chockler and Halpern, 2014 - https://www.aaai.org/Papers/JAIR/Vol22/JAIR-2204.pdf). The extended causal model could also solve this problem.

The idea of blame is also touched upon in Example 7. If Victoria knew she was able to rescue Betty without breaking her arm ($P = 2$) and chose to break her arm while rescuing her ($P = 1$), she caused Betty harm, because this goes against social norms. However, if she did not know she'd be able to do so and accidentally broke her arm ($P = 2$ was not an option), it is hard to argue that she caused Betty harm, as attempting to rescue Betty is well within social norms, and Betty's utility increased overall.

## Causing a decrease in utility in a socially unacceptable way (violating a default) is sufficient but not necessary for harm to occur

This criticism is aimed at the use of the "default utility" to represent all norms.

The work does not allow for situations where harm is done but reparations are made, and the agent in question is overall better off. For example:

> A thief assaults and robs a passerby but later, out of a guilty conscience, gives the money back, adding some additional money to more-than compensate for any physical and emotional distress caused. The passerby is happier than they would have been otherwise, because they needed the money.

While the passerby is better off by the end of the story, the thief did cause harm to the passerby in the intermediate step of assaulting and robbing them. While this issue *could* be addressed by stating that the default world is the world where the thief does not assault/rob and gives the passerby money for no reason, this seems to be a difficult argument to make, as the thief is expected by social norms only to not commit any crimes. Setting to default world to the world where the thief gives money to the passerby would mean that *everyone* is harming the passerby by not giving money to them.

An even more clear-cut version of this example follows:

> A thief assaults and robs a passerby. After the local media reports on the crimes committed, many people donate money to the passerby in sympathy. The passerby agrees that he is better off now than he would have been had he not been robbed and assaulted.

It seems that the thief's actions, contrary to his expectations, helped the passerby overall. Is it still true that the thief did not cause harm to the passerby?

The issue could also be fixed by using the extended causal model to, regardless of utility, label any world where the thief commits a crime as less normal than a world where the thief does not commit a crime.

**Strengths And Weaknesses:**

**Originality**: The paper's main idea of introducing a causal definition of harm follows (Bontly, 2016). The novelty of the method lies in the fact that the definition introduced is formally stated using Halpern and Pearl's language of causal models. ~However, the novel components that the new definition introduces – default utility + the observation that preventing a worse outcome does not constitute harm – do not seem original, as they seem to be subsumed by the extended causal model for the purpose of determining whether or not harm was caused. (Halpern and Hitchcock, 2011 - https://arxiv.org/pdf/1309.1226.pdf). The concepts of the default world and utility function could be naturally included in the partial preorder if the previous definition of extended causal models were used. This would mean that harm may be defined as modified Halpern-Pearl causation under the extended model, using a particular partial preorder. Perhaps the authors could comment on and clarify this point, in case I am missing any subtlety here.~ (retracted, see "Reply to reviewer's first response", parts 1 and 2 below)

**Quality**: ~The solution to the proposed research question ignores or misses prior work, as discussed above, and while~ While the solution to the proposed research question resolves an issue posed to a previous attempt in the literature made at defining harm using causation, the proposed definition lacks sufficient justification. ~Why would we need this specific definition instead of using the already existent one? Further, there~ There are no theoretical results introduced that demonstrate that counterexamples do not exist, and neither extensive human evaluation of the definition. Some potential counterexamples to the proposed definition are provided in the limitations section of this review.

**Clarity**: Overall, the paper is clearly written. The motivation of the paper is clear, as is the proposed definition, the example formulation, and the comparison of the definition's evaluations on the examples to those of the prior work mentioned in the paper.

**Significance**: Formalizations of harm become increasingly important as automated systems grow in ubiquity. Thus, the problem addressed by the paper and the idea of using actual causation to formalize the problem are important to the responsible development of large-scale AI systems. Overall, the motivation behind the paper is greatly appreciated.

---

> ### Author Response · Authors · 2022-08-01
> **Reply to reviewer: Part 2 of 2**
>
> Replies to questions:
>
> 4.
> By "qualitative" we just meant that our definition says that
> either there is harm or there isn't.  A quantitative definition might
> say that the degree of harm is, say, 0.3.  Of course, if our
> quantitative definition says that there is no harm, we would expect the
> (quantitative) degree of harm to be 0.  But if our definition says that
> there is harm, determining the degree of harm can be quite subtle (as
> we hinted at in the second paragraph of our conclusion).  We think that
> defining a useful quantitative notion of harm is an important topic, one
> that we are currently working on.  We though that the issues of
> quantitative harm were sufficiently subtle to deserve a separate paper.
>
> 5.
> Indeed, a partial order would suffice for the purposes of this
> paper. However, having a utility function
> mapping outcomes to numerical values is useful for the work in
> progress on quantification and aggregation of harm.
>
> 6.
> With regard to the reviewer's counterexamples, for the first example, we
> would say that the driver indeed harmed the resident, although presumably
> the driver should not be blamed for having done so, given that the
> resident put himself in a precarious position.  As we argued above, we
> believe it is important to separate harm and blame.  (The law makes a
> similar distinction.)  The same is true even if the resident
> deliberately jumped off the balcony upon hearing the driver's car
> honk.  In this case, of course, we are even less likely to blame the
> driver for the harm caused.
>
> For the second example, we would argue that the thief did harm the
> passerby by assaulting and robbing the passerby, even though the
> passerby is better off financially in the end. This is because there
> is more to the outcome (and hence the user's utility depends on more)
> than just how much money he has at the end.  The robbery and assault by
> itself has a negative utility.  (We believe that most people would
> agree, although we haven't conducted any experiments.)  With this
> choice of outcome, taking a reasonable utility function and taking the
> default utility to be that of not being robbed and not getting any money,
> our causal model would say that there is harm in this case.  This
> seems to us to be the "right" answer, and we are getting it for
> what seem to us the "right" reasons.
> (Of course, if the agent's utility function is such that the extra
> money compensates for the distress of being robbed and assaulted,
> then there is no harm.  We do not see a problem with that, although we
> suspect that this would not in fact be most people's utility function.)

---

> > ### Comment · Reviewer_6SwV · 2022-08-06
> > **Reply to (authors' rebuttal, part 2 of 2)**
> >
> > Thank you for clarifying the answers to questions 4 and 5 and for noting that for the purposes of this definition, causing harm is distinct from being responsible for harm.
> >
> > # Counterexample 1 (blame and harm)
> >
> > I believe that the paper would be strengthened by noting this distinction, as in the colloquial usage of the term "harm", the two are not as distinct. Particularly, could the authors provide references supporting the claim that "The law makes a similar distinction [between causing harm and blame]." applies to the second counterexample in this section? (for instance, the relevant portion of a reference supporting the claim that a court would rule that the driver caused the resident harm, despite the fact that the resident chose to jump)
> >
> > # Counterexample 2 (utility need not decrease for harm to occur)
> >
> > Regarding the second counterexample, my understanding of the authors' response is: "The premise (utility function) of the example is unrealistic. Even if the premise is realistic, there are no issues with our definition's evaluation that there is no harm in the counterexample." I disagree with both claims, and I think that this is an example of how human evaluation of the definition on provided examples (mentioned in the first limitation) could strengthen the paper.
> >
> > Regarding the second claim (that the definition correctly evaluates to "no harm" in the provided example), it is clear that the thief caused harm to the rights and physical well-being of the passerby, under the definition of harm in law - "loss of or damage to a person's right, property, or physical or mental well-being" (https://www.merriam-webster.com/dictionary/harm#legalDictionary) - because the passerby was robbed (violation of rights) and assaulted (physical damage). This holds regardless of the passerby's preference in terms of overall outcome (utility); even if the passerby's utility increases, they were harmed by the thief. What would be the authors' justification for this disparity between this legal definition of harm (which, in this case, I believe coincides with colloquial usage of "harm") and the definition proposed in the paper?
> >
> > Regarding the first claim (that the counterexample is unrealistic), I pose an isomorphic example that the authors may consider more realistic:
> >
> > > A rock falls from a height and gives a hiker a badly bruised arm, forcing her to stop hiking. While recovering on the trail, the hiker makes a life-long friend, whom she would not have met otherwise. Overall, the hiker thinks that getting hit by the rock was worth it, because though she spent a few weeks in a cast, she serendipitiously made a good friend; that is, if she had the choice, she would absolutely choose to meet her now-friend and bruise her arm again, rather than to stop the rock from bruising her arm and not meet her friend.
> >
> > It seems very strange to claim that the rock did not cause harm to the hiker because the overall utility of the hiker increased as a result of the rock hitting her. It is clear that the rock caused physical harm to the hiker under the definition of harm in law, because the hiker's body was damaged.
> >
> > If the authors still find any part of this example unrealistic, including the utility function based on the hiker's preferences, I will leave it to the authors to construct, or demonstrate it is impossible to construct, a realistic example where an event which on its own entails a negative change in utility (and is the cause of some injury, legally speaking) causes an event which entails a larger positive change in utility, such that the agent prefers both events over neither (a generalization of "situations where harm is done but reparations are made, and the agent in question is overall better off", per the review provided above).
> >
> > Finally, to confirm we share the same understanding of the utility function: the ordering of worlds induced by the utility function is consistent with the preferences of the agent in question, correct? That is, if the agent prefers one world (settings of variables which the agent observes) to another, then the former world's utility must be higher than the latter's, correct? If this is not the case, I think it should be noted and discussed in the paper, as this is distinct from typical use of the term "utility" in economics/game theory.

---

> > > ### Author Response · Authors · 2022-08-07
> > > **Reply to reviewer's second response**
> > >
> > > Response to discussion of Counterexample 1:
> > >
> > > > I believe that the paper would be strengthened by noting this distinction, as in the colloquial usage of the term "harm", the two are not as distinct.
> > >
> > > We are happy to follow your suggestion here.
> > >
> > > > Particularly, could the authors provide references supporting the claim that "The law makes a similar distinction [between causing harm and blame]." applies to the second counterexample in this section? (for instance, the relevant portion of a reference supporting the claim that a court would rule that the driver caused the resident harm, despite the fact that the resident chose to jump)
> > >
> > > We're not sure that there is a reference that deals with your particular example!  (We're not aware of one, in any case.)  But for
> > > the general point about distinguishing harm and blame, see https://en.wikipedia.org/wiki/Fault_(law) “Fault, as a legal term,
> > > refers to legal blameworthiness and responsibility in each area of law. It refers to both the actus reus and the mental state of the
> > > defendant. The basic principle is that a defendant should be able to contemplate the harm that his actions may cause, and therefore should
> > > aim to avoid such actions.”   We hope that this makes clear that, in the law, there is a strict separation between the mental state of the
> > > agent causing harm, and the notion of causing harm itself, and that the former is relevant only for establishing blame.
> > >
> > > Response to discussion of Counterexample 2:
> > >
> > > First, we agree that the example is realistic (and interesting). Second, we agree with your intuitions that the rock caused (physical)
> > > harm to hiker.  We believe we can capture what is going on in a reasonable way in our framework, we hope that you agree.  Consider two
> > > models.  In the first, the outcome involves only the hiker's physical condition (and, in particular, does not involve the fact that the
> > > hiker made a good friend).  In this model, with obvious choices of utility, the rock caused harm.  In the second model, the outcome
> > > includes the friendship, and there was no harm.  It seems to us that the first model captures what the law would do; it would deem the
> > > friendship to be irrelevant to the case, so would not include it in the outcome.  (We are not lawyers, so this claim should be taken with
> > > a grain of salt.  But even if the law doesn't do this, then people certainly do.  Not including the friendship in the outcome just
> > > formalizes the intuition that we are focusing on the physical damage caused by the rock.) The second model captures the "bigger picture".
> > > For us, harm is relative to a model (just like causality).  With your permission, we would be happy to include this example in the paper.
> > >
> > > > Finally, to confirm we share the same understanding of the utility function: the ordering of worlds induced by the utility function is consistent with the preferences of the agent in question, correct? That is, if the agent prefers one world (settings of variables which the agent observes) to another, then the former world's utility must be higher than the latter's, correct?
> > >
> > > Yes; this is exactly what we have in mind.  We are definitely thinking of utility in the way that game/decision theorists do.

---

> ### Author Response · Authors · 2022-08-01
> **Reply to reviewer: Part 1 of 2**
>
> While there is a sense in which the notion of default utility shares some common intuition with the notion of normality considered by Halpern and Hitchcock -- intuitively, we can think of the default utility as the utility of the most normal outcome -- there is no obvious sense in which we can embed the notion of harm in the extended causal models of Halpern and Hitchcock. To start with, extended causal models have no notion of utility. Of course, we could add utility, and define it on outcome variables just as we do in our paper, but this added structure plays no role in the definition of causality considered by Halpern and Hitchcock. Going on, things would be easier if we could assume that each variable had a unique default setting, as Hall (2007) for example does, because if the default value of O was o, we could take the default utility to be the utility of O=o. But Halpern and Hitchcock explicitly reject this assumption, and take the normality on worlds to be partial. Moreover, it is not clear what the default utility should be if there are several maximally normal worlds with outcomes that have different utilities. To make matters even worse, there is no need for the default utility in our paper to be the utility of a particular outcome; it could, for example, be the expected value of the utility of a number of outcomes.
>
> The preceding paragraph can be viewed as an argument that the "default world and utility function could be naturally included in the partial preorder" is not true. (As an aside, we do not have a "default world"; we have a default utility. This distinction is important.) It is also worth mentioning that in H2, while we use causality, and we can use extended causal models to define causality, the use of the normality ordering in doing so would be completely orthogonal to our use of default utility. Finally, we note that doing what the reviewer suggested, namely, trying to define harm in extended causal models using a particular partial preorder, would not suffice. If we were really going to take seriously using extended causal models as the basis of our definition, we would have to show how to define harm in an arbitrary extended causal model, perhaps further extended with a utility function. We hope that we have convinced the reviewer that this would be quite a nontrivial task.
>
> On a more general note, we believe our definition of harm is novel. Nothing in the Halpern and Hitchcock paper (a paper that we are very familiar with) hints at a definition of harm based on causal models, nor does any other paper that we are aware of, with the exception of RBT (as discussed in our paper).
>
> Replies to questions:
>
> 1 . Please see our response above - this is a new definition.
>
> 2. While we could presumably give a definition of harm that does not involve causality, we do not think it would deal well with any of the problematic examples in the literature. We wish we could prove a result showing that a good definition of harm must involve causality, but it's not clear what the formal statement of such a result would be. The same phenomenon arises with causality: it is hard to prove that a given definition is the "right" definition. What is done is to show how well the definition works on examples, especially ones that have been problematic for other definitions. This is exactly what we tried to do.
>
> 3. We did indeed think about the connection between harm and blame (and responsibility). We deliberately decided to view these as distinct notions. There are a number of reasons for this: First, there are obvious examples of harm that have nothing to do with blame, such as hurricanes. Second, more generally, while blame is obviously a moral notion, harm need not be. Blame (at least according to the Chockler-Halpern definition) is relative to the agent's mental state, as the reviewer mentions (specifically, the agent's beliefs, including beliefs about intentions and the effects of actions).
> For harm, at least according to our definition, no such mental model is required: in almost all circumstances, if you kill someone, you harm them, even if this was completely unintentional, and even if you could not have anticipated this result. Please see the discussion in Bradley’s paper (citation [3]), for a discussion of why an analysis of harm should be careful to leave out moral connotations.
> Note though that the mental model of the agent being harmed -- as opposed to an agent causing the harm -- does play some role in our definition, since we use the former agent's utility function.

---

> > ### Comment · Reviewer_6SwV · 2022-08-06
> > **Reply to (authors' rebuttal, part 1 of 2)**
> >
> > Thank you for clarifying your thoughts regarding the role of causation and blame in determining harm.
> >
> > # Question 3
> >
> > I think it is important to note the distinction between harm as defined in the paper and blame/responsibility, because the question of legal responsibility is mentioned in the abstract and introduction as a motivation for the paper.
> >
> > # Question 1
> >
> > > there is no obvious sense in which we can embed the notion of harm in the extended causal models of Halpern and Hitchcock.
> >
> > The sense suggested in the review above was to, for any particular causal model, choose a partial preorder consistent with the three criteria introduced in the authors’ definition of harm, given a utility function and default utility, and to define harm as causation within the extended model defined by this preorder and the original model. I am curious if there is a reason this is undesirable and that criteria on the partial order of utility + default utility must instead be used. In addition, I am curious about the necessity of default utility itself, as addressed in my second counterexample. Ultimately, I do not see any portion of the response to question 1 that addresses either curiosity.
> >
> > > To start with, extended causal models have no notion of utility. [...]
> >
> > I agree. However, neither point addresses my curiosity about why this particular structure of utility is necessary to define harm, rather than a partial preorder, especially given that the structure appears to be equivalent to a partial order (as the authors acknowledge in response to question 5).
> >
> > > Going on, things would be easier [...] But Halpern and Hitchcock explicitly reject this assumption, and take the normality on worlds to be partial.
> >
> > I do not see how the claims made in the sentences above imply that the concepts of the utility function or default utility are original or necessary (that is, that they cannot be incorporated into a partial order). Why would things be easier if each variable had a default setting? What is the issue with a partial preorder on worlds? Doesn’t rejecting this assumption mean that the extended causal model is more general?
> >
> > > Moreover, it is not clear what the default utility should be if there are several maximally normal worlds with outcomes that have different utilities.
> >
> > I agree. This seems to be an argument against the usage of default utility, not against the usage of the extended causal model, because it highlights a situation that default utility is not expressive enough to capture. To counter this argument, the burden would be on the authors to demonstrate that the situation described is unrealistic or impossible.
> >
> > > To make matters even worse, there is no need for the default utility in our paper to be the utility of a particular outcome; it could, for example, be the expected value of the utility of a number of outcomes. [...] (As an aside, we do not have a "default world"; we have a default utility. This distinction is important.)
> >
> > I agree that default utility need not be the utility of a particular world. However, setting default utility to a value in $[0, 1]$ that is not the default utility of a world is equivalent to setting it to the utility of the lowest-utility world which has utility higher than the default utility; thus, the default utility may be captured as the utility of this particular world. Why is the distinction between the default utility and the default world important, for the purposes of determining whether or not harm occurred; that is, why does this fact “make matters even worse”?
> >
> > > The preceding paragraph can be viewed as an argument that the "default world and utility function could be naturally included in the partial preorder" is not true.
> >
> > I do not see how the three points addressed above imply that it is not true that "default world and utility function could be naturally included in the partial preorder" or that the default world/utility function are original and necessary, rather than restrictions of the definition of the extended causal model.
> >
> > > Finally, we note that doing what the reviewer suggested, namely, trying to define harm in extended causal models using a particular partial preorder, would not suffice. [...]
> >
> > The review does not suggest defining harm *in* extended causal models using a particular preorder. Please see the above description for what it does suggest.

---

> > > ### Author Response · Authors · 2022-08-07
> > > **Reply to reviewer's first response, Part 2 of 2**
> > >
> > > > However, neither point addresses my curiosity about why this particular structure of utility is necessary to define harm, rather than a partial preorder, especially given that the structure appears to be equivalent to a partial order (as the authors acknowledge in response to question 5).
> > >
> > > Given that in our framework there is a single outcome variable O over which we have utilities, using a partial preorder on worlds rather than on the values of O seems to add little benefit, although one could certainly do so if one wishes to connect our work more closely to the HH framework. However, as we said, once we move to a more quantitative notion of harm (which we view as a necessary next step), having a quantitative utility (rather than a partial preorder) seems essential.
> > >
> > > >> Going on, things would be easier [...] But Halpern and Hitchcock explicitly reject this assumption, and take the normality on worlds to be partial.
> > > > I do not see how the claims made in the sentences above imply that the concepts of the utility function or default utility are original or necessary (that is, that they cannot be incorporated into a partial order).
> > >
> > > That wasn't the point we were trying to make with the sentences above. We were trying to argue that it was difficult to define harm in arbitrary extended causal models (which is what we thought you were claiming could/should be done).  We're sorry if we misunderstood your point.
> > >
> > > > Why would things be easier if each variable had a default setting? What is the issue with a partial preorder on worlds?
> > >
> > > At the risk of repeating ourselves, we certainly have no problem with a partial preorder on worlds, and agree that it could be used to capture some aspects of our definition (as shown above).
> > >
> > > > Doesn’t rejecting this assumption mean that the extended causal model is more general?
> > >
> > > We do not make this assumption either. Rather, we were simply considering it in order to go along with the reviewer’s proposal to use the partial ordering to somehow capture the default utility and utility function.
> > >
> > > In any case, as we pointed out, extended causal models are *not* more general than the
> > > models we use; since they cannot capture our definition of harm.
> > >
> > > >> Moreover, it is not clear what the default utility should be if there are several maximally normal worlds with outcomes that have different utilities.
> > > > I agree. This seems to be an argument against the usage of default utility, not against the usage of the extended causal model, because it highlights a situation that default utility is not expressive enough to capture. To counter this argument, the burden would be on the authors to demonstrate that the situation described is unrealistic or impossible.
> > >
> > > Default utility was never intended to capture the normality ordering on
> > > worlds!  As we noted above, the partial preorder that we would need to capture our
> > > definition may have nothing to do with the ordering imposed by normality
> > > considerations in the sense of HH (which is what extended causal models were intended to
> > > capture).   When we said "it is not clear" in this part of our
> > > response, we were referring to the exercise in which  “ we … think of
> > > the default utility as the utility of the most normal outcome”.  This
> > > is an exercise we carried out in order to go along with what we
> > > understood your proposal to be.  Nothing in our framework requires
> > > that we think of the default utility in this way. In fact, since for
> > > us the default  utility is simply an additional element of our model,
> > > this situation presents no problem for us whatsoever. We mentioned
> > > this situation merely to put pressure on the idea that the partial
> > > normality ordering could somehow be used to determine the default
> > > utility.
> > >
> > > > is equivalent to setting it to the utility of the lowest-utility world which has utility higher than the default utility
> > >
> > > But there need not be any world that has utility higher than the default utility; we do not require there to be one.
> > >
> > > > why does this fact “make matters even worse”?
> > >
> > > As before, it makes matters worse for the idea that the partial normality ordering in an arbitrary extended model could somehow be used to determine the default utility.
> > >
> > > >  it is not true that "default world and utility function could be naturally included in the partial preorder"
> > >
> > > In our reply, we examined what we take to be the natural method of including the default utility and utility into the partial preorder, and that method failed. As mentioned above, perhaps there does exist some more complicated method, which would be an interesting result.
> > >
> > > > The review does not suggest defining harm in extended causal models using a particular preorder. Please see the above description for what it does suggest.
> > >
> > > The above description suggests that we “define harm as causation within the extended model defined by this preorder and the original model.” This is simply what we meant.

---

> > > ### Author Response · Authors · 2022-08-07
> > > **Reply to reviewer's first response, Part 1 of 2**
> > >
> > > We thank the reviewer for their continued engagement with our paper.
> > >
> > > Response to discussion of Q3:
> > >
> > > We will do so.
> > >
> > > Response to discussion of Q1:
> > >
> > > > The sense suggested in the review above was to, for any particular causal model, choose a  partial preorder consistent with the three criteria introduced in the authors’ definition of harm, given a utility function and default utility, and to define harm as causation within the extended model defined by this preorder and the original model.
> > >
> > > What we tried to make clear, is that it is not at all obvious what it would mean to “choose a partial preorder consistent with…” We will explain this in a bit more detail now. The three criteria H1 – H3 do not express conditions on worlds, but on the utilities of outcomes in worlds that are in some way related to causation: H1 expresses a condition on (the utility of the outcome of) the actual world, H2 expresses a condition on the relation between the actual world and the counterfactual world that is used as a witness to show causation for some contrast value x’, and H3 expresses a condition on the relation between the actual world and the counterfactual world that results from the intervention do(X=x’) on the actual world. A partial preorder on worlds, on the other hand, simply expresses conditions on worlds themselves.
> > >
> > > What we take the reviewer to be saying, is that one can construct a mapping from a utility function on outcomes (together with a default utility) and a partial preordering on worlds that contain outcomes in such a manner that our conditions H1-H3 can be replaced by causation in the extended model. The most natural way to do so (but of course not the only way), would be to let the ordering be identical to the utility of its outcome, with the added property that all utilities which are at least as high as the default are mutually incomparable. Then one can define causation in extended causal models in the manner of HH. We believe we can prove that this definition would be equivalent to the combination of conditions H1 and H2. So although it does go some way towards translating our approach into the framework of HH, it doesn’t go far enough: it leaves out H3. Let us call this definition harm-HH, to distinguish it from our definition of harm.
> > >
> > > Perhaps there exists a more elaborate construction that also takes into account H3, we certainly did not prove yet that there is not. Note though that it will have to do more than simply focus on the ordering of the utilities, for the following reason. Imagine that H1 and H2 are satisfied but H3 is not, so that there is no harm. The HH-harm definition should now say that X=x rather than X=x’ is not a cause of O=o rather than O=o’ (where o’ is the outcome used in H2). Given that H2 is satisfied, the partial ordering should be such that the witness used for H2, whatever it is, is not as normal as the actual world. All we know about the witness used for H2 is that its utility is higher than that of the actual world, and that the witness W=w involves at least one variable (the latter is true because otherwise o’=o’’ and H3 would be satisfied). Given that the original idea was that higher utility certainly does not decrease the normality ordering, the most reasonable option is to make use only of the latter property. But in what way? If we impose the requirement that witnesses involving at least one variable are always less normal than the actual world, then causation reduces to but-for causation, which is certainly not an acceptable result. We hope we have convinced the reviewer that there is no obvious way in which our definition of harm can be expressed using causation in extended causal models.  Furthermore,  even ignoring H3, the approach suggested above leads to a specific subclass of extended models that is poorly motivated (unless you read our current paper), with a normality ordering that, roughly speaking, corresponds to utility. The normality ordering needed in this translation may clash with the one that we would use to capture the intuitions of HH, where normality capture intuitions like conventions, statistical frequency, or moral considerations.
> > >
> > > > I am curious about the necessity of default utility itself, as addressed in my second counterexample.
> > >
> > > We believe that a default utility is natural, and we seem to need it to deal with examples like the Batman example discussed in our paper (so that we can, for example, capture the distinction between a setting where Robin has no reason to expect golf clubs and one where he has been getting golf clubs from Batman every year, and thus has a reasonable expectation of getting them this year).

---

### Review · Ethics_Reviewer_ypWA · 2022-08-08

**Recommendation:** No changes are required.

**Ethics Review:**

This seems to be a false alarm.

---

### Review · Ethics_Reviewer_nbYt · 2022-08-18

**Recommendation:** No concerns on my part.

**Ethics Review:**

The reviewer raised a flag that a vaguely defined utility function could lead to ethical issues in practice. While this is true, I don't see it as an ethical issue with the work in this submission itself.

---

### Meta-Review · Area_Chair_ZFrP · 2022-08-26

**Recommendation:** Accept
**Confidence:** Certain

**Metareview:**

The paper aims at formally defining a qualitative notion of harm based on "actual causality". This is an important problem tackled, also or in even in particular for ML research. Many ethical issues raised about AI circle around a notion of harm, such as discrimination induced by classifiers, decision of autonomous agents, and even just harmful content of training sets. From this perspective, I really congratulate the authors. However, it is a downside that existing approaches to related approaches within the ML community are not discussed.  It is good tradition and practice to provide related work. Indeed, there is no actual causality model of harm yet (as far as I can say) but there are e.g. deontological approaches that could be mentioned or even briefly discussed such as

Liwei Jiang, Jena D. Hwang, Chandra Bhagavatula, Ronan Le Bras, Maxwell Forbes, Jon Borchardt, Jenny Liang, Oren Etzioni, Maarten Sap, Yejin Choi: Delphi: Towards Machine Ethics and Norms. CoRR abs/2110.07574 (2021)

Patrick Schramowski, Christopher Tauchmann, Kristian Kersting: Can Machines Help Us Answering Question 16 in Datasheets, and In Turn Reflecting on Inappropriate Content? In Proceedings of the ACM Conference on Fairness, Accountability, and Transparency (2022)

Patrick Schramowski, Cigdem Turan, Nico Andersen, Constantin A. Rothkopf, Kristian Kersting. Large pre-trained language models contain human-like biases of what is right and wrong to do. Nature Machine Intelligence 4(3): 258-268 (2022)

Moreover, the authors should discuss "we define an event to cause harm whenever it causes the utility of the outcome to be lower than the default utility" in more detail. Why is this a good definition? Let us for a moment equate utility with money then even too much money could be harmful due to network effects. In other words, it would be great if the authors could discuss a bit more the assumption made and their implications. This is even more important given the closeness to Bountly's notion of harm, replacing “state of affairs” by “outcomes”, and associating with each outcome a utility. The main downside, however, is the missing illustration of how the presented formalization of harm actually will help to tackle some of the ethical issues of AI. Is this really the right way? But then, who knows, and this paper is indeed taking a very different step than the average NeurIPS paper. In my opinion, the reviewers raise some salient arguments about the suitability of this paper for NeruIPS. For instance, there are indeed positive examples presented only. However, then the negative examples in the review want to illustrate that actual causality is not the right tool either. Still, the discussion within the NeurIPS community would then help already. Moreover, not knowing to break the arm is still causing harm. Anyhow, it is exactly this type of discussion that makes the paper in my humble opinion interesting to the NeurIPS community. The discussions with the authors showed that it is a topic that will provoke a lot of discussions. And since the overall sentiment of the reviewers is positive, I recommend accepting the paper.

**Award:**

No

---

### Decision · Program_Chairs · 2022-09-14

Accept